# SAYNEXT: A BENCHMARK AND COGNITIVELY IN-SPIRED FRAMEWORK FOR NEXT-UTTERANCE PREDICTION WITH MULTIMODAL LLMS

## ABSTRACT

We explore the use of large language models (LLMs) for next-utterance prediction in human dialogue. Despite recent advances in LLMs demonstrating their ability to engage in natural conversations with users, we show that even leading models surprisingly struggle to predict a human speaker's next utterance. Instead, humans can readily anticipate forthcoming utterances based on multi-modal cues—such as gestures, gaze, and emotional tone—from the context. To systematically examine whether LLMs can reproduce this ability, we propose **SayNext-Bench**, a benchmark that evaluates LLMs and Multimodal LLMs (MLLMs) on anticipating context-conditioned responses from multimodal cues spanning a variety of real-world scenarios. To support this benchmark, we build **SayNext-PC**, a novel large-scale dataset containing dialogues with rich multimodal cues. Building on this, we further develop a dual-route prediction MLLM, **SayNext-Chat**, that incorporates cognitive-inspired design to emulate the predictive processing in conversation. Experimental results demonstrate that our model outperforms state-of-the-art MLLMs in terms of lexical overlap, semantic similarity, and emotion consistency. Our results verify the feasibility of next-utterance prediction with LLMs from multimodal cues, and emphasize the indispensable role of non-verbal cues as the foundation of natural human interaction. We believe this exploration not only opens a new direction toward more human-like, context-sensitive AI interaction but also offers a pathway to uncovering cognitive concepts from dialogue data for human-centered AI. Our benchmark and model can be accessed at https://saynext.github.io/.

## 1 INTRODUCTION

Large language models (LLMs) have fundamentally reshaped the landscape of human–machine interaction. By leveraging large-scale pretraining and powerful generative capabilities, these models can produce coherent, contextually appropriate, and often engaging conversations with users (Brown et al., 2020). This unprecedented fluency has fueled a wide range of applications—from virtual assistants (OpenAI, 2022; Team et al., 2023; Liu et al., 2024) to educational tools (Wollny et al., 2021)—where LLMs are expected to act as adaptive and collaborative partners in dialogue.

Despite these advances, a fundamental paradox remains unresolved: while LLMs excel at sustaining dialogue, they consistently fail to accurately predict what a specific user will say next. We conducted preliminary experiments on several state-of-the-art conversational models, including GPT4o (Hurst et al., 2024), Gemini2.5 (Comanici et al., 2025), VideoLLaMA3 (Zhang et al., 2025), InternVL2 (Chen et al., 2024), LLaVA-NeXT (Zhang et al., 2024), and InstructBLIP (Dai et al., 2023), by prompting each model to predict a speaker's forthcoming utterance in a video, given contextual input and task instructions. Our results indicate that, even with explicit task instructions and visual context, all of these state-of-the-art models struggle to generate semantically appropriate predictions of forthcoming conversational utterances (See Figure 1).

This shortcoming is not a trivial flaw; rather, it reflects a fundamental limitation of LLMs. From a cognitive neuroscience perspective, next-utterance anticipation are central to natural communication, which involves *actively* generating hierarchical predictions about forthcoming sensory and

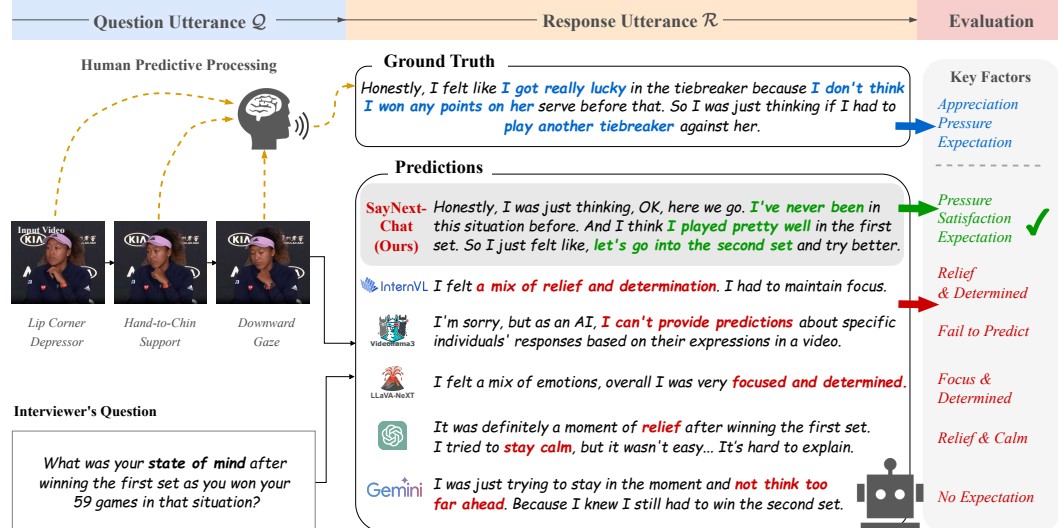

Figure 1: **The Illustration of Next-Utterance Prediction.** Given a video and the interviewer's question turn, the task in our benchmark is to predict the subsequent response. In evaluation, we show that the response utterance predicted by our model SayNext-Chat (green) align with those in the ground truth (blue), whereas other MLLMs (red) often yield irrelevant factors, incorrect ones, or entirely fail to predict. We extract the key factors of the predictions to facilitate readers' understanding. *More quantitative results are available and detailed in Sec. 4.1.*

linguistic input and continuously updating them against prediction errors (Rao & Ballard, 1999; Goldstein et al., 2022). Current LLMs, however, are *passively* optimized for statistical next-token prediction over large pretraining corpora—capturing general linguistic regularities, lacking the goal-directed, hierarchical, and intention-sensitive nature of human predictive processing. As a result, they fail to account for the idiosyncratic, personalized, and intentional aspects of human dialogue.

Another fundamental limitation of current large language models (LLMs) for next utterance prediction is that they rely primarily on textual content, thereby overlooking non-verbal cues that are central to human interaction. Non-verbal behaviors involve rich emotional cues and are the very basis on which humans anticipate and coordinate turns in conversation, serving as critical early indicators of communicative intent (Holler, 2025; Emmendorfer & Holler, 2025). Ignoring these modalities thus leaves LLMs blind to essential aspects of natural dialogue, making the modeling of non-verbal information indispensable for predictive and embodied language understanding.

A natural question arises: *What are the motivations for enabling LLMs to predict next utterances?* Apart from purely scientific interests, learning to predict forthcoming utterances carries broad implications across natural language processing, human–computer interaction, and AI safety. For example, in social robotics and embodied AI, such predictive capability supports more natural co-ordination with humans for fluid interaction in real-world environments. From the perspective of AI alignment and safety, next-utterance prediction also provides a principled way to model human intent in advance, thereby reducing the risk of misinterpretation and enabling proactive safeguards against harmful conversational trajectories. Ultimately, developing models that can anticipate dia-logue in a manner closer to human predictive processing is not only scientifically compelling but also practically essential for building trustworthy and interactive AI systems.

Motivated by these observations, we present the SAYNEXT and position it as an initial step to-ward bridging the gap between fluent dialogue generation and genuine cognitive understanding in human–AI interaction. By explicitly targeting next-utterance prediction, we move beyond the next-token paradigm of current LLMs and ground anticipation in multimodal cues that mirror human predictive processing. To this end, we introduce a dedicated dataset in two scales, SayNext-PC2K and SayNext-PC19K, of multimodal dialogues spanning real-world scenarios, and propose a novel MLLM framework, SayNext-Chat, that integrates verbal and non-verbal signals to infer user intentions and forecast forthcoming utterances. Our experiments across diverse conversational settings demonstrate the limitations of existing LLM models and the promise of our approach. More broadly, this work establishes a foundation for developing AI systems that can not only converse, but also

anticipate—thereby aligning more closely with the mechanisms of human communication and advancing the design of empathetic, adaptive, and trustworthy interactive agents.

Our contributions include:

1. We introduce **SayNext-Bench**, a multimodal benchmark designed to assess LLMs' capacity to integrate non-verbal visual cues for human-like predictive processing in conversational interaction, thereby laying a foundation for cognitively grounded dialogue modeling.

2. We construct the **SayNext-PC** dataset in two scales, PC2K and PC19K, through a scalable data collection pipeline. Extensive cross-scale and cross-domain evaluations verify the benchmark's extensibility and generalization ability.

3. We propose a cognitively inspired dual-route framework, **SayNext-Chat**, that incorporates priming vectors as learnable tokens, enabling LLMs to simulate anticipatory activation and align response generation with contextual priors.

4. We conduct systematic evaluations across four evaluation protocols and three metrics dimensions—lexical overlap, semantic similarity, and emotion consistency—revealing both the challenges and opportunities of predictive dialogue, and demonstrating that our model substantially outperforms state-of-the-art baselines.

## 2 SAYNEXT-BENCH

### 2.1 TASK SETUPS

In this work, a *conversational turn* is defined as a paired unit of discourse, consisting of an interlocutor's utterance and a subject's temporally contiguous response, which jointly serve as the fundamental granularity for prediction and evaluation. The SAYNEXT task is then formulated as predicting the forthcoming utterance $\tilde{T}_B^{\mathcal{R}}$ of the subject, conditioned on the interlocutor's preceding utterance $T_A^{\mathcal{Q}}$ and the subject's concurrent non-verbal expressions $V_B^{\mathcal{Q}}$.

The prediction problem can be expressed as approximating the conditional distribution of real responses $T_B^{\mathcal{R}}$ via an end-to-end mapping function $f_\theta$:

$$P\left(\tilde{T}_B^{\mathcal{R}} \mid T_A^{\mathcal{Q}}, V_B^{\mathcal{Q}}\right) \approx P\left(T_B^{\mathcal{R}} \mid T_A^{\mathcal{Q}}, V_B^{\mathcal{Q}}\right), \qquad f_\theta : (T_A^{\mathcal{Q}}, V_B^{\mathcal{Q}}) \mapsto \tilde{T}_B^{\mathcal{R}}. \tag{1}$$

### 2.2 THE SAYNEXT-PC DATASET

Based on the task definition, we find that no existing dataset is suitable for our task. Specifically, an eligible dataset must meet three essential criteria: (1) dyadic interaction, involving either two humans or one human and one machine; (2) multimodality, with at least synchronized video and text modalities; (3) continuous viewpoint, where the camera remains steadily focused on one interlocutor throughout the interaction. Frequent camera shifts or interruptions impede the reliable capture of facial and bodily expressions that are critical for inferring communicative intent and emotion. A detailed dataset investigation is provided in Appendix B.

To instantiate these principles, we introduce **SayNext-PC2K**, a novel dataset for next-utterance prediction, inspired by the post-match press conference setting in the iMiGUE dataset (Liu et al., 2021). For scalability, we further expand it to **SayNext-PC19K**, augmenting 259 original recordings to 3,463 videos from multiple tournaments. Both datasets capture athlete–interviewer dialogues with stable camera views, ensuring clear speech and expressive non-verbal cues.

Finally, the SayNext-PC2K comprises 2,092 minutes of dialogue spanning 5,432 turns, while SayNext-PC19K contains 20,766 minutes with 38,540 turns. Each dialogue is segmented into video clips and transcribed using Whisper (Radford et al., 2023). The transcription quality was verified, yielding an average Word Error Rate (WER) of 4.11%, which is within the range generally considered human-acceptable (see Appendix E for details).

### 2.3 EVALUATION PROTOCOL

We assess SayNext-Bench using four complementary protocols:

**Subject-Dependent Evaluation.** (Results in Sec. 4.2.2) In this setting, instances from the same subject in SayNext-PC2K appear in both the training and test sets. Conversational turns are randomly partitioned such that each subject in the test set has at least one corresponding instance in the training set, enabling the model to capture individual-specific nuances in emotional and behavioral expression. The split follows a 4:1 ratio between training and test data, and is fixed the same for all models.

**Subject-Independent Evaluation.** (Results in Sec. 4.2.2) In this setting, SayNext-PC2K is split such that subjects in the training and test sets form disjoint groups; that is, all subjects in the test set are unseen during training. The dataset comprises 72 subjects, divided into training and test groups at a 4:1 ratio. To mitigate potential cultural bias, subjects from five continents (North America, Australia, Europe, Asia, and Africa) are evenly represented across both groups.

**Cross-Scenario Evaluation.** (Results in Sec. 4.2.4) To examine robustness across conversational contexts, we adapt the IEMOCAP dataset (Busso et al., 2008), a dyadic motion-capture corpus comprising 12 hours of scripted and improvised dyadic dialogues. The dataset spans diverse scenarios such as job interviews, relationship conflicts, and casual exchanges, from which 4,113 conversational turns are extracted for our task. This protocol is to verify whether trained models can *directly* generalize from the solely press-conference scenario to different conversational settings in a zero-shot manner.

**Scalability Evaluation.** (Results in Sec. 4.2.4) To assess the robustness at large scales of the model performance, we construct SayNext-PC19K, a substantially larger dataset that extends the temporal span of conversations and incorporates broader cultural diversity. This broader scope is specifically designed to avoid potential overfitting issues with local optima in the SayNext-PC2K, thereby enhancing the capacity of SayNext-Bench to evaluate models under more diverse and fine-grained conditions.

## 3 THE PROPOSED METHOD

Inspired by a cognitive neuroscience perspective, we propose a dual-route prediction framework, SayNext-Chat, to anticipate forthcoming utterances, which incorporates learnable priming tokens representing the high-level belief priors from visual inputs and low-level cues directly perceived from multimodal inputs. An overview of the framework is shown in Figure 2 (a). We first introduce the generation of the priming factor, and then describe its role within the dual-route framework.

### 3.1 PRIMING FACTOR

In cognitive psychology, the *semantic priming* refers to the phenomenon that prior exposure to a stimulus facilitates subsequent processing of related information, typically occurring unconsciously and with minimal cognitive cost (Meyer & Schvaneveldt, 1971; Neely, 1977). Closely related to predictive processing, semantic priming suggests that the human brain pre-activates contextually relevant representations (often interpreted as prior beliefs) before speech, thus reducing the uncertainty of forthcoming utterances (Neely, 2012; Clark, 2013).

Following this observation, we leverage an LLM (GPT-4.1[1] in our implementation) to uncover recurrent, domain-relevant semantic associations from the training split only, which are then embedded and clustered into coherent latent structures. Each cluster is distilled by the LLM into a concise **priming factor**, constrained to be a neutral, widely recognized phrase denoting a specific behavioral or psychological characteristic. Along with the factor label, the LLM specifies a factor-specific polarity schema—**priming codebook**—that defines the meanings of its positive and negative poles. We then assign every response with a **target priming vector**, corresponding to a set of priming factors generated by LLM. The value of each entry in the priming vector lies in $[-1, 1]$: the magnitude encodes activation strength, and the sign encodes factor-specific polarity ($+1$ = strong positive manifestation, $-1$ = strong negative manifestation, $0$ = not manifested). In this way, the target priming vector serves as an auxiliary representation that *actively* encodes contextual prior beliefs about forthcoming utterances, reflecting the notion of pre-activation in predictive processing.

---

[1]Model card: gpt-4.1-2025-04-14

(a) The SayNext-Chat Framework.

(b) The Priming codebook of SayNext-PC2K.

| Priming factor | Range meaning in $[-1, 1]$ |
|---|---|
| Perceived Pressure | [low $\rightarrow$ high pressure] |
| Affect | [negative $\rightarrow$ positive] |
| Opponent Appraisal | [dismissive $\rightarrow$ respectful] |
| Motivation | [weak $\rightarrow$ strong drive] |
| Self-Efficacy | [doubtful $\rightarrow$ confident] |
| Self-Evaluation | [critical $\rightarrow$ positive] |
| Expectation Management | [disappointed $\rightarrow$ optimistic] |
| Physical State | [fatigued $\rightarrow$ healthy] |
| Acceptance | [resistant $\rightarrow$ accepting] |
| Adaptability | [rigid $\rightarrow$ flexible] |
| Uncertainty | [confident $\rightarrow$ uncertain] |
| Cognitive Stability | [unstable $\rightarrow$ stable] |
| Recovery Appraisal | [doubtful $\rightarrow$ optimistic] |
| Challenge Appraisal | [low $\rightarrow$ high challenge ] |
| Mental Focus | [distracted $\rightarrow$ focused] |
| Achievement Orientation | [indifferent $\rightarrow$ ambitious] |
| Social Connectedness | [isolated $\rightarrow$ connected] |
| Confidence | [doubtful $\rightarrow$ confident] |
| Self-Assurance | [anxious $\rightarrow$ assured] |
| Preparedness | [unprepared $\rightarrow$ prepared] |

Figure 2: (a) **The SayNext-Chat Framework.** (1) Priming factors are extracted through LLM-assisted induction to construct a priming codebook. (2) The codebook guides the LLM in assigning a target priming vector to each response. (3) During end-to-end training, the loss combines the MSE between target and predicted priming vectors with the cross-entropy loss from the LLM backbone. (b) **The Priming codebook of SayNext-PC2K.** Each entry contains a priming factor name, its explanation, and the meaning of range $[-1, 1]$. *Generation details are provided in Appendix F.*

## 3.2 SAYNEXT-CHAT: DUAL-ROUTE PREDICTION FRAMEWORK

Inspired by dual-route accounts of human cognition and the role of priors (Kahneman, 2011; Clark, 2013), we design SayNext-Chat with two complementary predictive routes: a **fast route** that directly maps low-level visual and textual cues to a response, and a **deep route** that infers high-level priors (priming factors) to guide generation.

**Fast route.** A visual backbone InternViT-300M, an MLP projector, and an LLM InternLM2.5-7B conditioned on prompts, text, and visual embeddings to produce the next utterance. **Deep route.** Non-verbal features are transformed by two convolutional layers and a fully connected layer into a *priming embedding*, injected into the LLM as a dedicated priming token (concatenated with visual tokens). In parallel, the embedding is mapped by another fully connected layer to a *priming vector*, supervised by the target priming vector from Sec. 3.1. Incorporating priming as learnable tokens improves output stability (Sec. 4.2.3) and aligns with hierarchical representations in cognition.

## 3.3 TRAINING STRATEGY

During training, both visual and language backbones are fine-tuned with Low-Rank Adaptation (LoRA) Hu et al. (2022) using rank $r = 16$, while the priming module is trained from scratch. The training objective consists of two components: the language modeling loss $\mathcal{L}_{\text{joint}}$, defined as the standard cross-entropy over tokens comparing predicted and ground-truth sequences, and the priming loss $\mathcal{L}_{\text{prim}}$, defined as an $\mathcal{L}_2$ regression encouraging the predicted priming vector to align with the target priming vector.

To balance optimization, we employ an *adaptive loss weighting* scheme. The overall training loss is

$$\mathcal{L} = \mathcal{L}_{\text{joint}} + \lambda_{\text{prim}} \mathcal{L}_{\text{prim}}, \tag{2}$$

where $\lambda_{\text{prim}}$ dynamically adjusts the relative importance of the priming loss. Specifically, it is updated according to an exponential moving average (EMA) of $\mathcal{L}_{\text{prim}}$:

$$\lambda_{\text{prim}} = \min\left(\max\left(\frac{\tilde{\mathcal{L}}_{\text{prim}}^{(t)}}{\mathcal{L}_{\text{prim}}^{(0)}+\epsilon}, 0\right), 1\right), \quad \tilde{\mathcal{L}}_{\text{prim}}^{(t)} = \mu\,\tilde{\mathcal{L}}_{\text{prim}}^{(t-1)} + (1-\mu)\,\mathcal{L}_{\text{prim}}^{(t)}. \tag{3}$$

Here, $\tilde{\mathcal{L}}_{\text{prim}}^{(t)}$ denotes the smoothed priming loss at step $t$. We set $\mu = 0.99$ and $\epsilon$ as a small constant for numerical stability. The training regimen employs AdamW optimizer with cosine learning rate decay initializing at $1 \times 10^{-4}$, and takes $2\times$A100 GPUs for training.

## 4 EXPERIMENTS

### 4.1 EVALUATION METRICS AND BASELINES

Evaluating next-utterance prediction is non-trivial. While word-for-word prediction would be the ideal outcome, such exact matching is nearly infeasible even for humans. What matters more is whether the underlying intent is correctly anticipated, with minor differences in wording and lexical choice being acceptable (see Fig. 1). This calls for a comprehensive evaluation methodology with a multi-faceted view of model performance. To this end, we carefully designed a three-dimensional evaluation framework encompassing six widely recognized metrics from NLP research. Sec. 4.2.5 also reports user studies and LLM-based evaluations to provide human-aligned perspectives.

First, **Lexical Overlap (LO)** is measured with BLEU-4 (Papineni et al., 2002) and ROUGE-L (Lin, 2004), which capture word- and phrase-level correspondence between predicted and reference responses. Second, **Semantic Similarity (SS)** is assessed using BERTScore-F1 (Zhang et al., 2020), which captures token-level semantic alignment, and Sentence-BERT (Reimers & Gurevych, 2019), which evaluates sentence-level embedding similarity. Third, **Emotion Consistency (EC)** is quantified by Valence and Arousal, computed via normalized profile comparisons derived from the NRC-VAD lexicon (Mohammad, 2018), thereby measuring affective congruence through emotion-bearing lexical matching. Details of metric computation are provided in Appendix D.

We evaluate SayNext-Chat against seven state-of-the-art multimodal LLM baselines, grouped into three categories: (1) frontier large-scale MLLMs, including GPT4o[2] and Gemini2.5-flash[3]; (2) open-source MLLMs with comparable parameter scales (7–8B) for fair comparison, including InternVL2-8B (Chen et al., 2024), VideoLLaMA3-7B (Zhang et al., 2025), LLaVA-NeXT-Video-7B (Zhang et al., 2024), and InstructBLIP-7B (Dai et al., 2023); and (3) task-specialized MLLMs for emotion recognition, represented by Emotion-LLaMA-7B (Cheng et al., 2024). All baselines are evaluated under identical prompts with temperature 0.7. Implementation details in Appendix C.

### 4.2 RESULTS AND ANALYSIS

#### 4.2.1 OVERALL TAKEAWAYS

- **Clear improvements with vision modality.** Results in Sec. 4.2.2 show that incorporating visual cues consistently improves next-utterance prediction performance.
- **SayNext-Chat outperforms baseline MLLMs.** Across all three evaluation dimensions, SayNext-Chat consistently surpasses zero-shot baselines, including frontier large-scale MLLMs, open-source models of comparable scale, and emotion-specific MLLMs (Sec. 4.2.2).
- **Priming vectors significantly boost emotional alignment.** While fine-tuning on domain-specific corpora increases both lexical overlap and semantic similarity, priming tokens further improve emotion accuracy of future utterances by 3% (Sec. 4.2.3).
- **Cross-scenario generalization and scalability.** SayNext-Chat maintains superior performance over compared baselines when evaluated on larger-scale datasets and across different scenarios in the zero-shot setting (Sec. 4.2.4).
- **Efficacy in human & LLM evaluations.** SayNext-Chat achieves higher scores in subjective human assessments, slightly surpassing GPT-4o and showing a clear margin over open-source MLLMs with comparable parameter scales (Sec. 4.2.5).

---

[2]Model card: gpt-4o

[3]Model card: Gemini 2.5 Flash

### 4.2.2 COMPARISON WITH STATE-OF-THE-ART METHODS

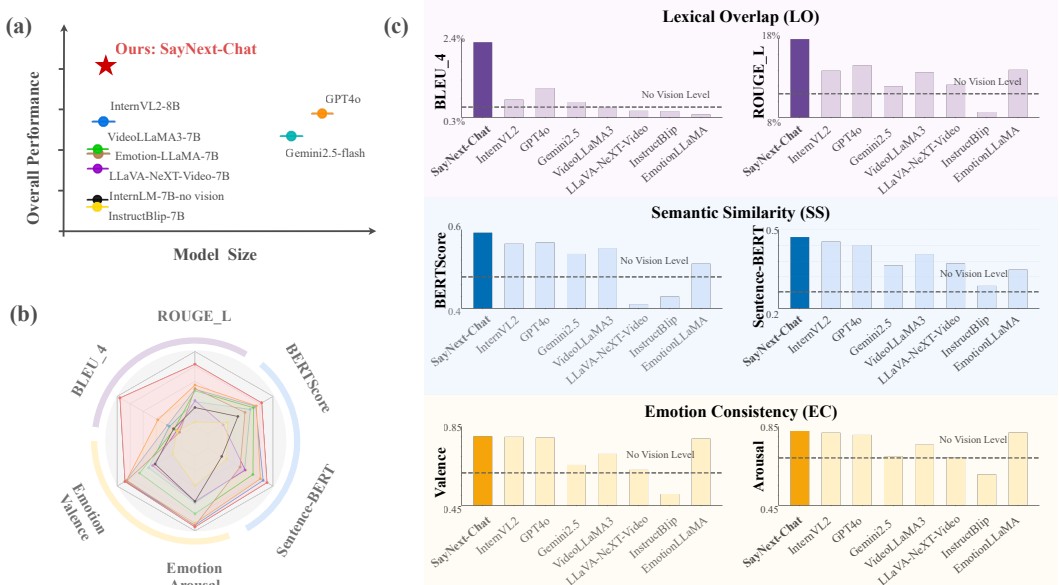

Figure 3: **Comparison of SayNext-Chat with State-of-the-art Baselines.** (a) Two-dimensional comparison of relative model size and performance (average rank across metrics). SayNext-Chat (red star) achieves both smaller size and higher performance. (b) Radar chart of multi-metric evaluation, where each polygon corresponds to a model and is colored consistently with (a). Our model (red) consistently ranks highest across all metrics in the radar chart. (c) Bar charts of six metrics across three dimensions. SayNext-Chat (the first bar) consistently attains the highest score.

We present the main results under the subject-dependent protocols in Figure 3. Figure 3 (a) reports the average rank of all models, showing that SayNext-Chat combines relatively small size with superior performance. Figure 3 (b) highlights notable discrepancies across evaluation dimensions: lexical overlap remains extremely low (below 20%), semantic similarity reaches a moderate level (around 0.4–0.6), and emotion consistency achieves the highest scores (around 0.7–0.8). These results underscore the inherent difficulty of reproducing word-level utterances, even with advanced fine-tuned MLLMs. To further illustrate model distinctions, Figure 3 (c) shows that SayNext-Chat consistently surpasses all state-of-the-art baselines across every metric. Specifically, despite considerable variability in phrasing, SayNext-Chat achieves a 2–6 fold improvement in lexical overlap over competing systems, ranks first in semantic similarity on both BERTScore and Sentence-BERT, and substantially improves emotion consistency by capturing latent connections between non-verbal cues and their linguistic realizations. Performance trends remain consistent across both subject-dependent and subject-independent protocols, though several metrics decrease slightly in the latter, reflecting the continued challenge of modeling subject-specific patterns. (Details in Appendix C.2)

Figure 4 presents case studies comparing priming-vector heatmaps with ground-truth and generated responses. High-score cases typically arise when the interviewer's question provides sufficient context, when salient non-verbal expressions enable affective alignment, and when the dialogue follows common patterns (e.g., joy after victory, pressure from an opponent). In contrast, low-score cases often involve off-topic noise and the inability of current MLLMs to reproduce human humor or linguistic creativity.

### 4.2.3 ABLATION STUDY OF THE LEARNABLE PRIMING TOKEN

To evaluate the role of priming information, we compare three variants: (1) a fine-tuned model without a priming module, (2) prompt-based models where priming factors or target priming vectors are inserted via prompt engineering, and (3) SayNext-Chat, which integrates the priming vector as a dedicated learnable token. Ablation experiments are conducted on the SayNext-PC2K dataset under the subject-dependent protocol.

Figure 4: **Case Study on SayNext-PC2K.** In high-score samples, the predicted priming vector heatmap closely matches the target. Red and Blue indicate positive and negative values in the priming vector, with corresponding highlights in the response text. Star, heart, and drop markers denote three representative priming factors, showing their alignment between predicted and target vectors (similar colors) and clarifying their semantic meaning (listed beneath the heatmap). Low-score samples exhibit special patterns in the target priming vector that are difficult to predict.

Table 1: Relative deltas (higher is better) w.r.t. *InternVL2-8B*. Priming improves all metrics and emotion consistency; fine-tuning degrades emotion alignment ability.

| Method | ΔBLEU-4/% | ΔROUGE-L/% | ΔBERTScore-F1 | ΔBERT sent. cos. | ΔEmot. Val. | ΔEmot. Arou. |
|---|---|---|---|---|---|---|
| Factor in Prompt | −0.708 | −3.343 | −0.03988 | −0.07260 | −0.10859 | −0.08465 |
| Vector in Prompt | −0.715 | −3.278 | −0.03517 | −0.06980 | −0.08949 | −0.08249 |
| Only Finetune | +1.308 | +3.664 | **+0.01870** | **+0.01770** | −0.02506 | −0.01751 |
| Finetune+Priming (Ours) | **+1.535** | **+4.021** | +0.01830 | +0.01760 | **+0.00277** | **+0.00878** |

Table 2: Relative deltas (higher is better) w.r.t. *VideoLLaMA3-7B*. Priming improves all metrics.

| Method | ΔBLEU-4/% | ΔROUGE-L/% | ΔBERTScore-F1 | ΔBERT sent. cos. | ΔEmot. Val. | ΔEmot. Arou. |
|---|---|---|---|---|---|---|
| Factor in Prompt | −0.425 | −2.189 | −0.01173 | −0.04620 | −0.01678 | −0.02802 |
| Vector in Prompt | −0.445 | −1.931 | −0.00570 | −0.05310 | −0.02608 | −0.04135 |
| Only Finetune | +1.309 | +3.586 | +0.02332 | +0.04560 | +0.06546 | +0.04309 |
| Finetune+Priming (Ours) | **+1.558** | **+3.816** | **+0.02501** | **+0.05930** | **+0.07069** | **+0.04652** |

As shown in Table 1, fine-tuning alone allows the model to acquire domain-relevant latent representations, but the incorporation of priming vectors further enhances its sensitivity to salient semantic cues, yielding gains of 2.8% in emotion valence and 2.6% in emotion arousal over the fine-tuned baseline. In contrast, prompt-based approaches perform poorly, as priming factors or vectors tend to divert the model toward analyzing their numerical values rather than focusing on the prediction task. The results in Table 2 show a similar trend on another MLLM—VideoLLaMA3, further indicating that our method consistently improves model performance and is effective beyond a single backbone. Overall, these findings highlight that stable integration of cognitive priors through learnable tokens is indispensable for modeling predictive dialogue.

### 4.2.4 CROSS-SCENARIO GENERALIZATION AND SCALABILITY

For cross-scenario evaluation, we adapt IEMOCAP to SayNext-Bench. As shown in Table 3, SayNext-Chat consistently outperforms all state-of-the-art zero-shot baselines, with only the arousal dimension slightly below EmotionLlama—an expected outcome given EmotionLlama's explicit emphasis on emotion. These results demonstrate the benchmark's extensibility across scenarios and the model's capacity for cross-dataset generalization. For scalability, as shown in Table 4, SayNext-Chat also achieves the best performance on the larger SayNext-PC19K dataset, despite its broader cultural diversity and extended temporal span. The overall metric distributions remain comparable to those observed on SayNext-PC2K, with only a slight decrease in emotion valence.

Additionally, a comparison of extracted priming factors across the three datasets reveals both scenario-specific patterns and stable recurring dimensions, such as *Affect*, *Confidence*, and *Disap-*

Table 3: Cross-scenario Evaluation Results on IEMOCAP.

| Model | LO /% | | SS | | EC | |
|---|---|---|---|---|---|---|
| | BLE.↑ | ROU.↑ | B-F1.↑ | B-sent.↑ | Val.↑ | Aro.↑ |
| InternVL2 | 0.44 | 9.60 | 0.50 | 0.18 | 0.51 | 0.57 |
| VideoLLaMA3 | 0.38 | 9.53 | 0.49 | 0.17 | 0.45 | 0.51 |
| LLaVA-NeXT | 0.44 | 8.21 | 0.49 | 0.15 | 0.39 | 0.47 |
| Emotion-LLaMA | 0.26 | 10.06 | 0.45 | 0.17 | 0.53 | **0.62** |
| GPT-4o | 0.47 | 9.50 | 0.47 | 0.18 | 0.44 | 0.50 |
| Gemini-2.5 | 0.91 | 9.13 | 0.50 | 0.15 | 0.44 | 0.50 |
| **SayNext-Chat** | **5.44** | **22.29** | **0.58** | **0.31** | **0.55** | 0.59 |

Table 4: Scalability Evaluation Results on SayNext-PC19K.

| Model | LO /% | | SS | | EC | |
|---|---|---|---|---|---|---|
| | BLE.↑ | ROU.↑ | B-F1.↑ | B-sent.↑ | Val.↑ | Aro.↑ |
| InternVL2 | 0.36 | 13.04 | 0.54 | 0.47 | **0.79** | 0.81 |
| VideoLLaMA3 | 0.18 | 11.91 | 0.52 | 0.41 | 0.73 | 0.77 |
| LLaVA-NeXT | 0.46 | 13.79 | 0.53 | 0.46 | 0.79 | 0.80 |
| Emotion-LLaMA | 0.25 | 12.66 | 0.50 | 0.36 | 0.79 | 0.81 |
| **SayNext-Chat** | **2.49** | **15.71** | **0.55** | **0.47** | 0.79 | **0.82** |

Table 5: User study results. SayNext-Chat achieves the highest best-selection rate across all three experimental groups. $E\langle N\rangle$ denotes the experimental group number.

| Model | E1 | E2 | E3 | Average |
|---|---|---|---|---|
| InternVL2 | 24.25% | 25.50% | 24.00% | 24.58% |
| Videollama3 | 17.50% | 13.50% | 11.75% | 14.25% |
| GPT-4o | 26.00% | 26.75% | 28.00% | 26.92% |
| **SayNext-Chat** | **32.25%** | **34.25%** | **36.25%** | **34.25%** |

Table 6: LLM evaluation results. SayNext-Chat attains the highest top-1 rate under both GPT-4.1 and Gemini2.5-Pro rankings.

| Model | GPT4.1 | Gemini2.5-pro | Average |
|---|---|---|---|
| InternVL2 | 21.78% | 25.19% | 23.48% |
| Videollama3 | 16.30% | 12.30% | 14.30% |
| EmotionLlama | 6.81% | 5.48% | 6.15% |
| LlaVA-NeXT | 17.93% | 15.70% | 16.81% |
| InstructBLIP | 7.26% | 9.33% | 8.30% |
| **SayNext-Chat** | **29.93%** | **32.00%** | **30.96%** |

*pointment*. This convergence suggests that certain cognitive–affective factors are broadly universal across conversational contexts. More broadly, it implies that scaling the size and diversity of datasets may support the development of a generalized inventory of priming vectors, potentially applicable to everyday dialogue and informative for cognitive modeling.

### 4.2.5 USER STUDY & LLM-AS-JUDGE EVALUATION

Beyond standard metrics, we run two subjective evaluations to capture human alignment and holistic preferences. In the user study, 12 participants (divided into three groups) repeatedly evaluated the same 400-question test set, selecting the candidate most similar to the reference from InternVL2, VideoLLaMA3, GPT-4o, and our model; outputs were randomized to mitigate order effects, and per-model selection rates appear in Table 5. For LLM-based judging, GPT-4.1 and Gemini2.5-Pro[4] ranked predictions from five baselines and our model; we exclude GPT-4o and Gemini to avoid RLHF-induced family bias, and report top-1 rates in Table 6.

The user study shows that SayNext-Chat achieves the highest best-selection rate (34.25%), ahead of GPT-4o (26.92%), while VideollaMA3 attains the lowest share despite a similar parameter scale. In the LLM evaluation, SayNext-Chat ranks first under both evaluators. These results indicate that SayNext-Chat outperforms strong baselines. Moreover, preference distributions are consistent across user-study groups and across LLM evaluators, suggesting robust inter-evaluator agreement.

## 5 LIMITATIONS & FUTURE WORKS

Although our experimental results on SayNext-Bench reveal the potential of LLMs to predict forthcoming utterances in human dialogue by leveraging multimodal signals, several limitations remain. First, the low lexical-overlap accuracy across all models highlights the intrinsic difficulty of the task and indicates the need for more cognitively capable MLLMs. Given both individual variation and the stochastic nature of language, incorporating multi-turn dialogue could help capture speaker-specific habits and background knowledge, thereby improving lexical alignment. Second, dataset quality imposes constraints: although existing resources adapt reasonably well to our task, noisy samples (e.g., the low-score example in Figure 4)—such as inconsistencies between speech and expression or off-topic conversations—can hinder accurate learning.

---

[4] Model card: Gemini-2.5-Pro

Another essential challenge of SAYNEXT lies in the **complexity of evaluation**. Next-utterance prediction inherently involves a broad set of cognitive and perceptual competencies, making it difficult to design metrics that fully capture the richness of human anticipatory processing. While we make a substantial effort to provide a comprehensive evaluation, covering wording-level, semantic-level, emotion-level, and human/LLM preference judgments, we acknowledge that these metrics still have their own limitations. For example, wording-level metrics cannot account for the one-to-many nature of open-ended prediction; semantic metrics depend on pretrained language encoders that may not fully reflect context-specific meaning; emotion metrics approximate latent affect using discrete categories or embeddings that imperfectly represent nuanced emotional states; and human/LLM preference judgments, while informative and more reliable, introduce reproducibility issue and are costly to scale. These limitations highlight the need for more cognitively grounded evaluation methods that can better capture the richness of human anticipatory processing.

**Position of this work**. At a high level, our task is not a conventional dialogue-prediction problem (i.e., next-token prediction). Instead, our goal is to investigate a fundamental limitation in current LLMs and MLLMs: although these models excel at sustaining dialogue in traditional NLP settings, they consistently fail to predict what a specific human will say next. This gap is nontrivial and reflects a deeper issue: LLMs operate as passive next-token generators rather than systems capable of engaging in human-like anticipatory cognition. SAYNEXT (next-utterance prediction) is therefore designed as **a proxy task** to model aspects of human predictive processing. Performing well on this task **implicitly** includes intention inference, affective anticipation, and multimodal social reasoning—capabilities that are broadly lacking in current LLMs. Consequently, we remind readers that the SAYNEXT task does not fit neatly into existing categories such as emotion recognition, intention classification, or dialogue continuation. Instead, it opens a distinct and complementary research direction that we hope will motivate future work in cognitively grounded predictive modeling and inspire next-generation LLMs with more human-like anticipatory abilities.

**Future directions**. Analyzing SAYNEXT by investigating more nuanced pragmatic–emotional expressions, such as sarcasm, humor, and metaphor, can be a promising direction, which remains challenging even for advanced LLMs. Another promising direction is integrating a multi-turn or richer multimodal context may help the model capture speaker-specific habits and discourse dynamics. Moreover, developing cognitively grounded evaluation frameworks that better capture intention alignment and anticipatory reasoning, potentially through collaboration with cognitive science and social psychology. Finally, analyzing the learned priming factors as cognitively meaningful dimensions offers an opportunity to build more interpretable and human-aligned anticipatory models.

## 6 CONCLUSION

We present SAYNEXT, a multimodal next-utterance prediction task for trustworthy, seamless human–AI dialogue, and release SayNext-Bench with the scalable SayNext-PC dataset spanning four evaluation protocols. Inspired by a cognitive neuroscience perspective, we propose SayNext-Chat, a dual-route framework that injects learnable priming tokens to fuse perceptual cues with anticipatory priors. Across subject-dependent/independent settings and larger, cross-scenario datasets, SayNext-Chat consistently surpasses strong baselines on lexical overlap, semantic similarity, and emotion consistency, corroborated by user studies and LLM-as-judge evaluations. Ablation studies show that the proposed priming factors and the incorporation of visual information materially improve prediction accuracy. Together, these findings validate the feasibility and exploratory potential of SAYNEXT and help bridge cognitively grounded mechanisms of human communication with the design of more empathetic, adaptive, and trustworthy interactive AI agents.

**Ethics Statement.** This study was conducted in compliance with the General Data Protection Regulation (GDPR); data collection, storage, and sharing followed legally compliant procedures. In the user study, we anonymized records and obtained informed consent from all evaluators. Data were sourced from publicly available media and licensed corpora (e.g., IEMOCAP), with appropriate citations. We provide access links only—without redistributing raw media. We also employed multiple evaluation protocols and datasets spanning different scales and scenarios to mitigate bias.

**Reproducibility Statement.** Our benchmark and models will be made available, and can be found at our website: `https://saynext.github.io/`.

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

APPENDIX CONTENTS

## A    LLM Usage Statement

Large Language Models (ChatGPT) were used exclusively to improve the clarity and fluency of English writing. They were not involved in research ideation, experimental design, data analysis, or interpretation. The authors take full responsibility for all content.

## B    Dataset

### B.1    Dataset Comparison

Although there are plenty of existing multimodal emotion recognition datasets (in Table.7), we find they all fail to be directly applied to our proposed task. For example, the widely used MELD dataset Poria et al. (2019) — composed of video segments cropped from television shows — suffers from frequent viewpoint changes, which impede the reliable capture of visual expressions of subjects for emotion inference. Recent datasets such as MaSaC Bedi et al. (2021) and CPED Chen et al. (2022) exhibit similar issues, which we refer to as the "*Interrupted*" view in this paper. Although earlier datasets like IEMOCAP Busso et al. (2008) and MSP-IMPROV Busso et al. (2016) offer consistent recording of subjects in conversational videos (we refer as "*Continuous*" recording view), they are limited by small sample sizes and a restricted number of speakers. The SEMAINE dataset McKeown et al. (2011), which records interactions between humans and virtual visual characters, is the closest match; however, due to language generation technology constraints in 2012, the verbal output from the virtual characters is ineffective, diminishing the meaning to predict natural language expression. To fill this gap and inspired by Liu et al. (2021), we introduce the *SayNext-PC* dataset, specifically designed for the *SayNext* benchmark. A comparative overview of *SayNext-PC*, scale in PC2K and PC19K, and other related datasets is presented in Table.7.

Table 7: **Attributes comparison between SayNext-PC and other widely used datasets for multimodal emotion recognition in dialogue.** Note: *#* denotes the number of instances; *(s/min)/(U/D)* indicates the average duration in seconds per utterance or per dialogue; *F/M* represents female/male; *A*: audio, *V*: video, *T*: text, *M*: dynamic motion capture, *Mi*: micro body expression.

| Dataset | #Dia. | #Utter. | Ave. Dur. | #Subj. (F/M) | Res. | Modality | Descriptive label | Continuous view |
|---|---|---|---|---|---|---|---|---|
| IEMOCAP | 5 | 10039 | 4.5s/U | 10 (5/5) | $720 \times 480$ | A/V/T/M | ✗ | ✓ |
| SEMAINE | 959 | 5816 | 5min/D | 150 | $720 \times 580$ | A/V/T | ✗ | ✓ |
| MSP-IMPROV | 6 | 7818 | 4s/U | 12 (6/6) | $1440 \times 1080$ | A/V | ✗ | ✓ |
| MELD | 1433 | 13708 | 3.59s/U | 260+ | $1280 \times 720$ | A/V/T | ✗ | ✗ |
| MaSaC | 1190 | 15576 | 20s/U | - | - | A/V/T | ✗ | ✗ |
| CPED | 12000 | 132762 | 2.1s/U | 392 | - | A/V/T | ✗ | ✗ |
| *SayNext-PC2K* | **359** | **5432** | **22.44s/U** | **72 (36/36)** | $\mathbf{1280 \times 720}$ | **A/V/T/Mi** | ✓ | ✓ |
| *SayNext-PC19K* | **3463** | **38540** | **30.368s/U** | **474 (246/228)** | $\mathbf{640 \times 360}$ | **A/V/T** | ✓ | ✓ |

### B.2    Dataset Settings

**The SayNext-PC2K dataset** comprising 359 post-match press conference videos from Grand Slam tournaments was collected from online platforms, yielding a total duration of 2,092 minutes. All videos were recorded at a resolution of $1280 \times 720$ and a frame rate of 25 fps. Micro-body language annotations were directly adopted as non-verbal visual labels. Speaker diarization was employed to segment the videos into question-answer pairs, resulting in 5,432 pairs with an average utterance duration of 22.44 seconds.

Based on two evaluation protocols discussed in the manuscript, two data split settings were employed: subject-dependent and subject-independent.

In the subject-dependent setting, samples were randomly assigned to the training and test sets, with careful partitioning to ensure that each subject appears in both sets. To achieve a 4:1 ratio, the training dataset comprises 2,041 samples, while the test dataset consists of 675 samples.

In the subject-independent evaluation, a comprehensive statistical analysis was performed to examine the distribution of subject identifiers and their corresponding nationalities. The training set comprises 58 unique subject IDs, representing a diverse spectrum of nationalities including:

- *Spain, Bulgaria, Russia, the United Kingdom, Switzerland, France, Germany, Belgium, Austria, the United States, Croatia, Canada, the Czech Republic, Australia, Denmark, Slovakia, South Korea, South Africa, Serbia, Argentina, Romania, Japan, Taiwan, Latvia, Greece, China, Italy.*

In contrast, the test set contains 14 unique subject IDs, with nationalities including:

- *Switzerland, Japan, the United Kingdom, Canada, Australia, Germany, Spain, Russia, Ukraine, the United States, China, South Africa.*

Notably, the test set spans five continents, thereby underscoring the cultural universality of the dataset. These statistics highlight the international diversity within both the training and test subsets, which in turn attests to the robustness and representativeness of our data. To achieve a 4:1 ratio, the training dataset comprises 2,155 samples, while the test dataset consists of 561 samples.

**The SayNext-PC19K dataset** comprises 3,463 post-match press conference videos collected from a wider spectrum of tournaments, including the Australian Open, US Open, and Wimbledon. Relative to SayNext-PC2K (2017–2019), the temporal coverage has been substantially extended to span 2017–2024, thereby encompassing more recent competitions and reflecting the evolving dynamics of professional tennis discourse. The number of participating athletes has also expanded markedly, from 72 to 474, which introduces cross-cultural characteristics and heterogeneous communicative norms and expressive tendencies. In terms of data quality, one quarter of the videos are available at a resolution of $1280 \times 720$, while the remainder are at $640 \times 360$. Each video is meticulously annotated with a unique athlete identifier and nationality, ensuring traceability across sessions and enabling systematic analysis of subject-level and cultural factors.

**The IEMOCAP dataset** is an interactive emotional dyadic motion-capture corpus collected by the Speech Analysis & Interpretation Laboratory at the University of Southern California. It consists of dyadic sessions in which 10 actors (5 female and 5 male) perform either improvisations or scripted scenarios, yielding approximately 12 hours of audiovisual data, including video, speech, facial motion capture, and text transcriptions. The sessions are manually segmented into utterances, each annotated by at least three human annotators. For our SAYNEXT task, we adopt the dataset based on its transcriptions, which have been shown to exhibit a low Word Error Rate (WER). Nevertheless, the utterances are much shorter in duration compared with SayNext-PC, reflecting the context-sparse nature of the IEMOCAP corpus and making next-utterance prediction in single-turn conversations particularly challenging.

To provide a clearer overview of its scenario diversity, we classify and enumerate the major dialogue scenarios in IEMOCAP in Table.8.

## C EXPERIMENT DETAILS

### C.1 EXPERIMENTAL SETUPS

**Video Input.** Videos are processed as sequences of frames, with up to 4 frames used during fine-tuning and up to 16 frames during inference. Frames are extracted at intervals of 8. Specifically, in fine-tuning, 4 frames are randomly sampled in each epoch using a sliding-window strategy, which provides dynamic visual information while reducing memory costs. For consistency, all models employing InternViT as the visual backbone (zero-shot baselines, fine-tuned models, and SAYNEXT) resize frames to $480 \times 480$.

**Inference Configuration.** For zero-shot baselines, we evaluate the SAYNEXT task using seven state-of-the-art MLLMs. Owing to the inherent complexity of these models, numerous hyperparameters and configuration settings must be considered during fine-tuning. To clarify our experimental setup, we highlight several important modifications. In InstructBLIP, only a single frame is used for prediction, as the model lacks a dedicated video modality input. Similarly, in EmotionLLaMA, we also utilize a single frame, since our experiments indicate that this configuration yields better performance. This improvement is likely attributable to the reduced redundancy in visual embed-

Table 8: Coarse-to-fine categorization of IEMOCAP dialogue scenarios

| Coarse Category | Fine-grained Scenario | Appears in |
|---|---|---|
| **Workplace / Administrative** | Customer–service conflict (e.g., ID office argument) | Improvised |
| | Bureaucratic frustration (forms, documents, procedures) | Improvised |
| | Clerk–customer misunderstanding | Improvised |
| | Workplace tension / poor service attitude | Improvised |
| **Casual / Light Conversation** | Neutral conversation before conflict escalation | Scripted |
| | Small-talk transitions embedded in scripted scenes | Scripted |
| | Everyday conversational prelude | Minimal presence |
| **Family / Domestic** | Family dispute about a missing or presumed-dead relative | Scripted |
| | Parent–child argument about responsibilities | Scripted |
| | Sibling disagreement / family stress | Scripted & Improvised |
| | Coping with grief inside the family | Scripted |
| **Romantic / Interpersonal** | Couple jealousy argument | Improvised |
| | Break-up confrontation | Improvised |
| | Suspicion or betrayal between partners | Improvised |
| | Reconciliation attempt | Improvised |
| **Supportive / Apology** | Apologizing after doing something wrong | Improvised |
| | Consoling a grieving friend | Improvised |
| | Comforting someone after a negative event | Improvised |
| | Emotional reassurance / empathetic support | Improvised |
| **General Conflict** | Accusation and denial | Improvised |
| | Heated disagreement between acquaintances | Scripted & Improvised |
| | Blaming someone for wrongdoing | Improvised |
| | Confronting someone about past actions | Scripted |
| **Miscellaneous (Scripted Drama)** | Classic theatre conflict (e.g., "Who's Afraid of Virginia Woolf?", "The Apartment") | Scripted |
| | Emotional monologues inside dramatic scenes | Scripted |
| | High-tension theatrical exchanges | Scripted |

dings that typically arises when processing multiple frames. VideoLLaMA3 and LLaVA-NeXT, by contrast, are provided with four input frames, consistent with our model. For GPT-4o and Gemini-2.5-Flash, we extract four frames for GPT-4o and adapt Gemini's native video interface to process visual information at approximately 1 fps.

For all baselines and our model, we apply the same prompt to generate predictions (full prompt in Appendix I). For InternLM-based inference, we adopt consistent generation settings as follows:

```
no_repeat_ngram_size=2,
repetition_penalty=1,
do_sample=True,
temperature=0.7,
top_k=50,
top_p=0.9,
max_new_tokens=2048
```

## C.2 FULL RESULT TABLES

We present the complete experimental results (tables and figures) in this section, providing numerical evidence for accurate reference.

First, Figure 5 shows the radar plots of the subject-dependent and subject-independent protocols. Both exhibit similar hexagonal patterns, where our model achieves the best overall performance. Across baselines, the no-vision model (black) forms the smallest hexagon, while GPT-4o and In-

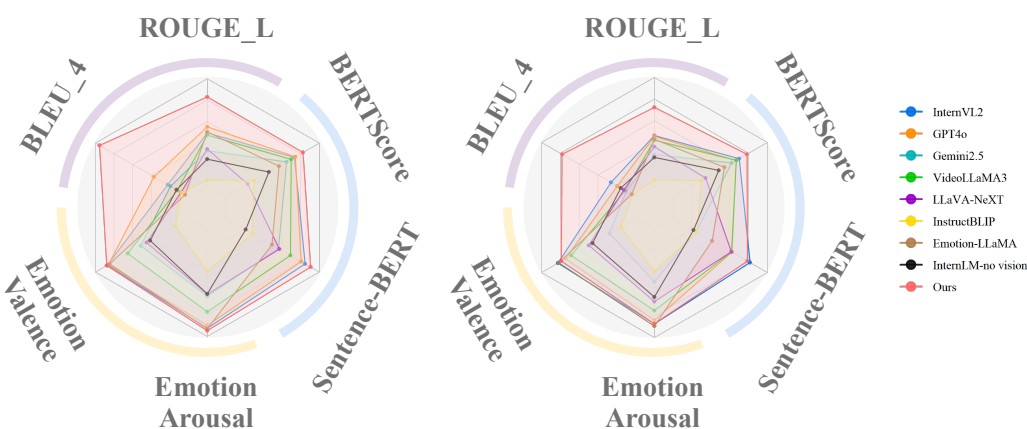

Figure 5: **Multidimensional comparison between baseline modales and our method in subject-dependent and subject-independent protocols.** Our model (red) forms the largest polygon in both subject-dependent and subject-independent settings, highlighting its outstanding performance in lexical overlap.

ternVL2 perform better than the remaining baselines. Notably, the hexagon of our model under the subject-independent protocol is slightly smaller than that under the subject-dependent setting, indicating that subject-independent evaluation is more challenging—likely due to the model's limited ability to capture individual-specific patterns.

The detailed quantitative results for both subject-dependent and subject-independent protocols are reported in the following two tables (Tables 9 and 10).

Table 9: **Numerical results on SayNext-PC2k on Subject-Dependent Evaluation Protocol.** Best results shown in **bold**, and the second-best is underlined. The proposed approach achieves superior performance across all metrics. Note: BLEU-4 and ROUGE-L values are in percentages.

| Method | Lexical Overlap /% | | Semantic Similarity | | Emotion Consistency | |
|---|---|---|---|---|---|---|
| | BLEU-4↑ | ROUGE-L↑ | BertScore-F1↑ | Sentence-BERT↑ | Valence↑ | Arousal↑ |
| No vision | 0.574 | 11.005 | 0.4921 | 0.2632 | 0.61549 | 0.69160 |
| InternVL2 | 0.772 | 13.936 | 0.5468 | 0.4546 | 0.79863 | 0.81969 |
| VideoLLaMA3 | 0.554 | 13.696 | 0.5397 | 0.4072 | 0.71166 | 0.75820 |
| LLaVA-NeXT | 0.472 | 12.143 | 0.4469 | 0.3715 | 0.63044 | 0.69315 |
| InstructBlip | 0.466 | 8.679 | 0.4596 | 0.2856 | 0.50761 | 0.60748 |
| Emotion-LLaMA | 0.380 | 14.046 | 0.5139 | 0.3478 | 0.78967 | 0.82126 |
| GPT4o | 1.081 | 14.623 | 0.5489 | 0.4415 | 0.79441 | 0.80838 |
| Gemini2.5 | 0.710 | 11.951 | 0.5300 | 0.3686 | 0.65538 | 0.70022 |
| SayNext-Chat (**Ours**) | **2.307** | **17.957** | **0.5651** | **0.4722** | **0.80140** | **0.82847** |

Table 11 reports the experimental results on IEMOCAP, demonstrating the transferability of our model across different contexts and scenarios.

Table 12 presents the experimental results on SayNext-PC19K, demonstrating the generalizability of our model to larger-scale datasets. The results further suggest potential scale effects in the SAYNEXT task and benchmark, indicating that training on sufficiently large and fine-grained corpora could foster the development of more generalizable next-utterance prediction models. Such models would lay the groundwork for more advanced conversational AI and for trustworthy, safe human–computer interaction, while also offering an AI-driven perspective on the mechanisms underlying human cognition.

Table 10: **Numerical results on SayNext-PC2k on Subject-Independent Evaluation Protocol.** Best results shown in **bold**, and the second-best is underlined. The proposed approach achieves superior performance across all metrics. Note: BLEU-4 and ROUGE-L values are in percentages.

| Method | Lexical Overlap /% | | Semantic Similarity | | Emotion Consistency | |
|---|---|---|---|---|---|---|
| | BLEU-4↑ | ROUGE-L↑ | BertScore-F1↑ | Sentence-BERT↑ | Valence↑ | Arousal↑ |
| No vision | 0.633 | 11.148 | 0.4968 | 0.2644 | 0.63313 | 0.70029 |
| InternVL2 | 0.850 | 13.576 | 0.5402 | **0.4447** | **0.78024** | 0.79932 |
| VideoLLaMA3 | 0.534 | 13.033 | 0.5339 | 0.3857 | 0.72401 | 0.75002 |
| LLaVA-NeXT | 0.565 | 12.332 | 0.4686 | 0.3852 | 0.64883 | 0.71677 |
| InstructBlip | 0.371 | 8.643 | 0.4596 | 0.2759 | 0.51285 | 0.60562 |
| Emotion-LLaMA | 0.389 | 13.144 | 0.5086 | 0.3240 | 0.77705 | **0.80732** |
| GPT4o | 0.710 | 13.470 | 0.5334 | 0.3872 | 0.74770 | 0.78584 |
| Gemini2.5 | 0.510 | 11.446 | 0.5244 | 0.2731 | 0.56094 | 0.64392 |
| SayNext-Chat (**Ours**) | **1.939** | **16.681** | **0.5563** | 0.4365 | 0.76580 | 0.79932 |

Table 11: **Numerical results on Cross-Scenarios Evaluation Protocol.** The train dataset and test dataset are both from adapted IEMOCAP dataset. Best results shown in **bold**, and the second-best is underlined. The proposed approach achieves superior performance across all metrics. Note: BLEU-4 and ROUGE-L values are in percentages.

| Method | Lexical Overlap /% | | Semantic Similarity | | Emotion Consistency | |
|---|---|---|---|---|---|---|
| | BLEU-4↑ | ROUGE-L↑ | BertScore-F1↑ | Sentence-BERT↑ | Valence↑ | Arousal↑ |
| InternVL2 | 0.438 | 9.602 | 0.5016 | 0.1804 | 0.51430 | 0.57000 |
| VideoLLaMA3 | 0.379 | 9.528 | 0.4915 | 0.1674 | 0.44701 | 0.51264 |
| LLaVA-NeXT | 0.164 | 7.047 | 0.4779 | 0.1724 | 0.39481 | 0.46541 |
| Emotion-LLaMA | 0.257 | 10.058 | 0.4478 | 0.1724 | 0.52532 | **0.61817** |
| GPT4o | 0.474 | 9.504 | 0.4707 | 0.1787 | 0.43655 | 0.49732 |
| Gemini2.5 | 0.914 | 9.126 | 0.4957 | 0.1528 | 0.43701 | 0.50057 |
| SayNext-Chat (**Ours**) | **5.439** | **22.292** | **0.57625** | **0.3076** | **0.54767** | 0.59014 |

Table 12: **Experimental results on SayNext-PC19K across different models.** Best results shown in **bold**, and the second-best is underlined. The proposed approach achieves superior performance across all metrics. Commercial models are not tested here due to high expense. Note: BLEU-4 and ROUGE-L values are in percentages.

| Method | Lexical Overlap /% | | Semantic Similarity | | Emotion Consistency | |
|---|---|---|---|---|---|---|
| | BLEU-4↑ | ROUGE-L↑ | BertScore-F1↑ | Sentence-BERT↑ | Valence↑ | Arousal↑ |
| InternVL2 | 0.358 | 13.042 | 0.5387 | 0.4679 | **0.79117** | 0.81085 |
| VideoLLaMA3 | 0.183 | 11.913 | 0.5181 | 0.4149 | 0.73393 | 0.76607 |
| LLaVA-NeXT | 0.455 | 13.792 | 0.5290 | 0.4593 | 0.78622 | 0.80323 |
| Emotion-LLaMA | 0.254 | 12.664 | 0.5023 | 0.3558 | 0.78894 | 0.81350 |
| SayNext-Chat (**Ours**) | **2.493** | **15.709** | **0.5482** | **0.4712** | 0.78623 | **0.81864** |

## C.3 ADDITIONAL CROSS-DATASET VALIDATION

To further verify that the observed performance trends are not due to adaptation artifacts, we perform additional cross-dataset validation under multiple training–testing combinations. Concretely, we consider: (i) training on our 2K subset and testing on IEMOCAP / 19K, (ii) training on IEMOCAP and testing on 2K / 19K, and (iii) training on 19K and testing on IEMOCAP. For each setting, we report all six evaluation metrics used in the main paper, covering lexical overlap, semantic similarity, and emotion consistency.

As summarized in Table 13, models trained on larger and more diverse datasets (e.g., 19K) consistently achieve stronger generalization across test sets, particularly in semantic similarity (BERTScore-F1 and Sentence-BERT) and emotion consistency (Valence, Arousal). In contrast, models trained on smaller datasets (2K or IEMOCAP alone) exhibit notable drops when transferred

Table 13: Cross-dataset validation under different training–testing combinations.

| Method | Lexical Overlap /% | | Semantic Similarity | | Emotion Consistency | |
|---|---|---|---|---|---|---|
| Training $\rightarrow$ Testing | BLEU-4↑ | ROUGE-L↑ | BERTScore-F1↑ | Sentence-BERT↑ | Valence↑ | Arousal↑ |
| 2K $\rightarrow$ IEMOCAP | 0.893 | 10.695 | 0.4925 | 0.1382 | 0.55709 | 0.62108 |
| 2K $\rightarrow$ 19K | 0.861 | 15.218 | 0.5475 | 0.4367 | 0.75789 | 0.78664 |
| IEMOCAP $\rightarrow$ 19K | 0.023 | 7.829 | 0.4644 | 0.1640 | 0.49756 | 0.60204 |
| IEMOCAP $\rightarrow$ 2K | 0.128 | 8.913 | 0.4747 | 0.1518 | 0.54655 | 0.63653 |
| 19K $\rightarrow$ IEMOCAP | 1.156 | 10.901 | 0.5077 | 0.1491 | 0.51188 | 0.56456 |
| 19K $\rightarrow$ 2K | 1.931 | 17.439 | 0.5655 | 0.4582 | 0.76968 | 0.79182 |

to more challenging test distributions. Overall, these results reinforce that the performance patterns follow dataset size and difficulty, rather than being driven by any specific adaptation or overfitting artifact.

# D  EVALUATION METRICS

## D.1  LEXICAL OVERLAP

Lexical-overlap metrics quantify surface-form similarity between a candidate text and a reference, and are widely used in NLP to assess content preservation and phrasing fidelity. Although they emphasize word- and phrase-level matches rather than deep semantics, they provide a reproducible, task-agnostic signal that complements embedding-based measures. In our setting, the reference is the real response transcribed by Whisper and the candidate is the model's prediction; unless otherwise noted, we report scores as percentages, with higher values indicating better overlap.

**BLEU-4.**  BLEU (Papineni et al., 2002) computes modified n-gram precisions up to 4-grams and penalizes overly short hypotheses via a brevity penalty. For a candidate $C$ and a set of references $\mathcal{R}$, the modified precision for order $n$ is

$$p_n = \frac{\sum_{g \in \text{ngrams}_n(C)} \min\bigl(\text{count}_C(g), \max_{r \in \mathcal{R}} \text{count}_r(g)\bigr)}{\sum_{g \in \text{ngrams}_n(C)} \text{count}_C(g)}, \tag{4}$$

where $C$ is the candidate (model output); $\mathcal{R}$ is the set of references; $g$ is an $n$-gram; $\text{ngrams}_n(C)$ is the multiset of $n$-grams extracted from $C$; $\text{count}_X(g)$ is the occurrence count of $g$ in text $X$.

The brevity penalty (BP) is

$$\text{BP} = \begin{cases} 1, & |C| > |r^*|, \\ \exp\bigl(1 - |r^*|/|C|\bigr), & |C| \le |r^*|, \end{cases} \tag{5}$$

where $|X|$ denotes the token length of text $X$; $r^* \in \mathcal{R}$ is the reference whose length is closest to $|C|$. BLEU-4 is then

$$\text{BLEU-4} = \text{BP} \cdot \exp\left(\sum_{n=1}^{4} w_n \log p_n\right), \qquad w_n = \tfrac{1}{4}, \tag{6}$$

where $w_n$ is the weight for $n$-gram order $n$ (uniform for BLEU-4).

We compute corpus-level BLEU-4 with standard smoothing for zero-count n-grams.

**ROUGE-L**  ROUGE-L (Lin, 2004) measures sequence-level overlap via the longest common sub-sequence (LCS), capturing in-order matches without requiring contiguity. Let $L$ be the LCS length between candidate $C$ and reference $R$. Define

$$R_{\text{LCS}} = \frac{L}{|R|}, \qquad P_{\text{LCS}} = \frac{L}{|C|}, \tag{7}$$

where $|\cdot|$ denotes token length, and the $F$-measure is obtained as follows,

$$\text{ROUGE-L} = \frac{(1 + \beta^2) R_{\text{LCS}} P_{\text{LCS}}}{R_{\text{LCS}} + \beta^2 P_{\text{LCS}}}, \tag{8}$$

where $\beta$ controls the recall–precision trade-off (we use $\beta = 1$ for $F_1$). We adopt the common $F_1$ variant ($\beta = 1$) and report the mean over the evaluation set.

## D.2 SEMANTIC SIMILARITY

Because lexical overlap does not reward legitimate paraphrases with divergent surface forms—especially in open-ended next-utterance prediction—we interpret BLEU-4 and ROUGE-L alongside semantic metrics to obtain a more faithful assessment of model behavior.

**BERTScore.**  We employ BERTScore Zhang et al. (2020) with the `deberta-xlarge -mnli` backbone to compute contextual embedding similarity:

$$R_{\text{BERT}} = \frac{1}{|x|} \sum_{x_i \in x} \max_{\hat{x}_j \in \hat{x}} \mathbf{x}_i^\top \hat{\mathbf{x}}_j, \tag{9}$$

$$P_{\text{BERT}} = \frac{1}{|\hat{x}|} \sum_{\hat{x}_j \in \hat{x}} \max_{x_i \in x} \mathbf{x}_i^\top \hat{\mathbf{x}}_j, \tag{10}$$

$$F_{\text{BERT}} = 2 \frac{R_{\text{BERT}} \cdot P_{\text{BERT}}}{R_{\text{BERT}} + P_{\text{BERT}}}, \tag{11}$$

where $x_i$ denotes reference token, $\hat{x}$ the candidate token, and $\mathbf{x}_i$, $\hat{\mathbf{x}}_j$ their respective contextual embeddings.

BERTScore computes token-level semantic similarity using contextual embeddings (via cosine similarity), producing precision, recall, and their harmonic mean $F_1$. In our setting, precision reflects how much of the *generated* content is semantically supported by the reference next utterance—high precision indicates the model avoids adding irrelevant or hallucinated material ("what it predicts is right"). Recall reflects how much of the *reference* content is covered by the generation—high recall indicates the model captures more of the salient information present in the gold response ("it includes more of what should be said"). Because next-utterance prediction requires both accuracy and coverage, we report BERTScore-$F_1$ as a balanced summary.

**Sentence-BERT.**  Unlike BERTScore, which aggregates token-level matches, we also assess holistic, sentence-level alignment using Sentence-BERT (SBERT) (Reimers & Gurevych, 2019) cosine similarity, serving as a complementary semantic metric.

$$\text{SBERT-cos}(C, R) = \frac{\mathbf{e}_C^\top \mathbf{e}_R}{\|\mathbf{e}_C\|_2 \|\mathbf{e}_R\|_2} \in [-1, 1], \tag{12}$$

where $C$ is the candidate (predicted next utterance); $R$ is the reference (real next utterance); $\mathbf{e}_C = f_{\text{SBERT}}(C) \in \mathbb{R}^d$ and $\mathbf{e}_R = f_{\text{SBERT}}(R)$ are their sentence embeddings; $f_{\text{SBERT}}(\cdot)$ denotes the Sentence-BERT encoder; $(\cdot)^\top$ is the dot product; $\|\cdot\|_2$ is the Euclidean norm. Larger values indicate stronger sentence-level semantic similarity. We report mean scores over the evaluation set.

## D.3 EMOTION CONSISTENCY

The affective alignment measurement leverages the NRC-VAD lexicon (Mohammad, 2018) containing 20,000 emotion-annotated lexical entries, where each term $\omega$ is quantified along four emotion dimensions: Valence (V, emotional positivity), Arousal (A, activation level), Dominance (D, control perception). For a reference-candidate text pair $(R, C)$, the alignment score is computed as:

$$S = 1 - \frac{1}{4} \sum_{k \in V, A, D} \left| \bar{s}_k^{(R)} - \bar{s}_k^{(C)} \right| - \beta \cdot |\rho_R - \rho_C|, \tag{13}$$

where $\bar{s}_k = \frac{1}{|W|} \sum_{\omega \in W} \hat{s}_k(\omega)$ denotes the normalized mean score of dimension $k$, with $\hat{s}_k(\omega)$ being the min-max scaled value of $s_k(\omega)$ from the lexicon. The lexical coverage ratio ($\rho_R$, $\rho_C$) measures

the proportion of lexicon-matched tokens, and $\beta = 0.8$ controls the penalty weight for coverage disparity.

Implementation proceeds through three phases: text preprocessing first tokenizes and lemmatizes inputs using EmotionDynamics. All dimension scores are then normalized across the lexicon's value range. Finally, dimension-wise averages and coverage ratios are computed for both texts, followed by the composite scoring in Eq. 13.

# E  WORD ERROR RATE (WER)

To ensure the reliability of automatic transcripts used for evaluation, we assess transcription quality with Word Error Rate (WER). We use Whisper to transcribe all videos in SAYNEXT-PC2K (332,651 words in total) and compute WER against human annotations. Formally,

$$\text{WER} = \frac{S + D + I}{N} \times 100\%, \tag{14}$$

where $S$, $D$, and $I$ denote the number of substitutions, deletions, and insertions, respectively, and $N$ is the number of words in the human reference.

As a challenging, representative case, we evaluate a non-native speaker (subject #42): the average WER is $4.11\%$ with a range from $0\%$ to $11.76\%$ across utterances. For comparison, the Whisper paper reports English WERs of $4.1\%$ and $9.3\%$ (Radford et al., 2023), and an oft-cited estimate of human WER is about $4\%$. These results indicate that our transcripts are accurate enough for downstream evaluation.

**Selection of Whisper model.**  We conducted preliminary transcription experiments across Whisper model sizes and selected the *medium* variant after validation. The *small* model underperformed on interview-style speech, while the *large* model tended to aggressively normalize by removing interjections (e.g., "um", "oh"), potentially erasing paralinguistic cues that are informative for emotional analysis.

# F  CODEBOOK & PRIMING FACTOR ANALYSIS

## F.1  CLUSTER EVALUATION

We sample $N{=}200$ training utterances in SayNext-PC2K, ensuring coverage of all subjects' responses. We first use GPT-4.1 to extract *basic factors* from this subset, then apply $k$-means to group the factors into clusters. Given the resulting clusters, GPT-4.1 induces a *priming codebook* by naming each factor, providing a concise explanation, and specifying the meaning of the value range $[-1, 1]$. Using this codebook, GPT-4.1 is subsequently guided to assign a consistent *target priming vector* to every reference response in the dataset.

Because clustering quality directly affects codebook induction, we quantify it using the *Silhouette Coefficient* (SC). For each item $i$ with representation $\mathbf{x}_i$, we define the within-cluster dissimilarity $a_i$ (lower is better) and across-cluster dissimilarity $b_i$ (higher is better) as:

$$a_i = \frac{1}{|C_i| - 1} \sum_{\substack{j \in C_i \\ j \neq i}} d(\mathbf{x}_i, \mathbf{x}_j), \qquad b_i = \min_{C \neq C_i} \frac{1}{|C|} \sum_{j \in C} d(\mathbf{x}_i, \mathbf{x}_j), \tag{15}$$

where $C_i$ is the cluster containing $i$; $C$ ranges over all other clusters; $d(\cdot, \cdot)$ is the Euclidean distance.

The silhouette $s_i$ summarizes separation vs. cohesion for item $i$. It is defined as

$$s_i = \frac{b_i - a_i}{\max\{a_i, b_i\}} \in [-1, 1]. \tag{16}$$

Accordingly, the overall score is the mean silhouette:

$$\text{SC} = \frac{1}{N} \sum_{i=1}^{N} s_i. \tag{17}$$

We evaluate $k \in \{10, 15, 20\}$ under three trimming settings $\tau \in \{0, 0.05, 0.10\}$. As shown in Figure 6, $k=20$ with $\tau=0.05$ attains the highest (trimmed) silhouette score; we therefore adopt $k=20$ for codebook induction. We did not explore larger $k$ because excessively many priming factors degraded the consistency of GPT-4.1 when assigning vector dimensions in practice (i.e., reduced stability beyond 20 dimensions). The full priming codebook is provided in the next section.

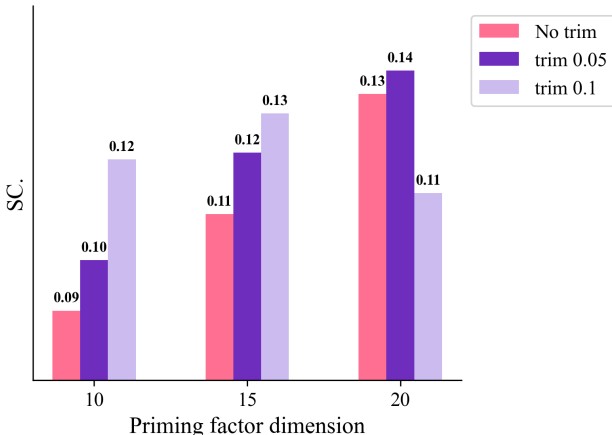

Figure 6: The Silhouette Coefficient (SC) value comparison of different settings of clustering.

## F.2 FULL CODEBOOKS

Table 14: Priming Codebook of SayNext-PC2K with 20 Priming Factors.

| | |
|---|---|
| Perceived Pressure | Reflects the player's subjective experience of psychological pressure during and after the match, indicating whether they felt burdened or at ease in high-stakes moments. |
| | 1 represents high or burdensome pressure (nervous, defensive), -1 represents low or relieved pressure (calm, comfortable). |
| Affect | Represents the overall positive or negative emotional tone expressed by the player, encompassing feelings such as joy, satisfaction, discomfort, or emotional struggle. |
| | 1 represents positive affect (joy, satisfaction), -1 represents negative affect (discomfort, struggle). |
| Opponent Appraisal | Reflects the player's evaluative judgment of the opponent's abilities, performance, and qualities, indicating a positive or negative assessment that shapes the tone and content of the interview language. |
| | 1 represents positive appraisal (admiration, respect), -1 represents negative appraisal (criticism, dismissal). |
| Motivation | Represents the player's drive, ambition, and determination to compete and succeed, as reflected in their language about goals, effort, and competitive intent. This factor captures the positive or negative intensity of their motivational state post-match. |
| | 1 represents high motivation (strong drive/ambition), -1 represents low motivation (lack of drive/ambition). |
| Self-Efficacy | Reflects the player's confidence in their ability to improve and achieve desired performance outcomes, encompassing both positive self-assessment and recognition of areas for growth. |
| | 1 represents high self-efficacy (confidence in improvement), -1 represents low self-efficacy (doubt or regret about improvement). |

| Self-Evaluation | Represents the player's cognitive and emotional assessment of their own performance, encompassing both positive and negative self-appraisal, self-criticism, and reflection on actions and outcomes. |
| --- | --- |
| | 1 represents positive self-evaluation (confidence, satisfaction), -1 represents negative self-evaluation (self-criticism, disappointment). |
| Expectation Management | Reflects the player's cognitive and emotional processing of outcomes relative to their prior expectations, encompassing surprise, disappointment, regret, optimism, and hopefulness about future events. |
| | 1 represents positive expectation management (optimism, hopefulness, positive anticipation), -1 represents negative expectation management (disappointment, regret, surprise at negative outcomes). |
| Physical State | Represents the player's self-reported physical condition, encompassing fatigue, discomfort, readiness, and overall bodily well-being, which influences their language and emotional tone in post-match interviews. |
| | 1 represents positive physical state (readiness, comfort), -1 represents negative physical state (fatigue, discomfort, limitation). |
| Acceptance | Reflects the player's cognitive and emotional acknowledgment of circumstances, setbacks, or changes, indicating a willingness to adapt or reconcile with outcomes, whether positive or negative. |
| | 1 represents high acceptance (open, adaptive), -1 represents low acceptance (resistant, avoidant). |
| Adaptability | Reflects the player's cognitive and emotional flexibility in response to changing circumstances, novel experiences, and evolving environments, as indicated by frequent references to learning, openness, adjustment, and recognition of change. |
| | 1 represents high adaptability (openness, learning, adjustment); -1 represents low adaptability (rigidity, discomfort with change). |
| Uncertainty | Reflects the player's cognitive and emotional response to unpredictability and ambiguity regarding match conditions, performance, and outcomes, influencing their language with expressions of doubt, surprise, or conditional statements. |
| | 1 represents high uncertainty (expressions of doubt, surprise, or unpredictability), -1 represents low uncertainty (expressions of certainty, confidence, or predictability). |
| Cognitive Stability | Reflects the player's perceived steadiness or fluctuation in thought processes and self-assessment, ranging from consistent, focused cognition to uncertainty and confusion. |
| | 1 represents cognitive stability (clarity, consistency), -1 represents cognitive instability (confusion, uncertainty). |
| Recovery Appraisal | Reflects the player's cognitive and emotional assessment of their physical and psychological recovery process, including optimism, relief, concern, and confidence regarding overcoming setbacks or injuries. |
| | 1 represents positive appraisal (optimism, relief, confidence in recovery), -1 represents negative appraisal (concern, doubt, ongoing struggle with recovery). |
| Challenge Appraisal | Represents the player's cognitive and emotional assessment of the degree and nature of challenges faced during the match, including tactical, technical, physical, and situational difficulties, as well as their perceived ability to cope with and adapt to these challenges. |

| | |
|---|---|
| | 1 represents high perceived challenge (player discusses significant obstacles and adaptation), -1 represents low perceived challenge (player reports minimal difficulty or smooth performance). |
| Mental Focus | Represents the degree to which the player's language centers on concentration, present-moment awareness, and cognitive engagement with the match, reflecting either a strong or disrupted mental focus. |
| | 1 represents strong mental focus and present-moment engagement, -1 represents disrupted or scattered mental focus. |
| Achievement Orientation | Reflects the player's focus on accomplishment, ambition, and the pursuit or recognition of significant goals, which shapes their emotional and cognitive responses in post-match interviews. This factor encompasses expressions of pride, satisfaction, motivation, and validation related to personal or collective achievements. |
| | 1 represents strong achievement focus (expressed pride, ambition, or satisfaction), -1 represents minimal achievement focus (lack of reference to accomplishment or ambition). |
| Social Connectedness | Reflects the player's sense of belonging, appreciation, and positive regard for others, including peers, mentors, audience, and support teams, indicating a positive emotional bias toward interpersonal relationships and communal support. |
| | 1 represents strong social connectedness (gratitude, admiration, appreciation), -1 represents weak or absent social connectedness (disdain, lack of respect). |
| Confidence | Reflects the player's cognitive and emotional appraisal of their own abilities, encompassing self-belief, determination, and receptiveness to encouragement or inspiration, which influences their post-match language in a positive or negative direction. |
| | 1 represents high confidence (assertive, self-assured), -1 represents low confidence (doubtful, apologetic). |
| Self-Assurance | Represents the player's internal sense of certainty and trust in their abilities, which influences their language and demeanor in post-match interviews. It reflects a positive or negative cognitive-emotional state regarding their own competence and readiness. |
| | 1 represents high self-assurance (expressed certainty, composure), -1 represents low self-assurance (expressed doubt, anxiety). |
| Preparedness | Reflects the player's perceived level of readiness and adequacy of preparation, encompassing both physical and mental aspects, which influences their confidence and outlook in post-match communication. |
| | 1 represents high preparedness (well-prepared), -1 represents low preparedness (unprepared). |

### F.3 ABLATION: CHOICE OF LLM FOR PRIMING-VECTOR GENERATION

We additionally include experiments using Llama-3.1 (open-source) and Gemini-2.5-Flash (low-cost) to generate priming vectors. The results are summarized in Table. 15.

Table 15: Comparison of different LLMs for priming-vector generation.

| Model | BLEU-4↑ | ROUGE-L↑ | BertScore-F1↑ | Sentence-BERT↑ | Valence↑ | Arousal↑ |
|---|---|---|---|---|---|---|
| SayNext-Chat (Llama-3.1) | 1.729 | 16.948 | 0.5619 | 0.4498 | 0.76999 | 0.80241 |
| SayNext-Chat (Gemini-2.5-flash) | 1.792 | 17.134 | 0.5621 | 0.4460 | 0.76816 | 0.79312 |
| SayNext-Chat (GPT-4.1) **(Ours)** | **2.307** | **17.957** | **0.5651** | **0.4722** | **0.80140** | **0.82847** |

The results demonstrate that using Llama-3.1 (open-source) or Gemini-2.5-Flash (low-cost) to generate priming vectors yields performance that is broadly comparable across semantic and wording-level metrics, and maintains stable trends on affect-related measures. This indicates that the SayNext framework does not rely on a specific proprietary model and remains functional with open or inexpensive alternatives. GPT-4.1, however, provides the most consistent improvements across all dimensions, and we therefore adopt it for constructing the final benchmark.

## G USER STUDY

To align with our goal of enabling seamless and trustworthy human–AI conversation, we conducted a human evaluation of next-utterance prediction. We implemented a web interface (Figure 7) in which, on each trial, participants viewed the reference answer, then selected the option *closest to the human-written reference answer* from four anonymized model outputs. The four candidates (InternVL-8B, VideoLLaMA3, GPT-4o, and SayNext-Chat) were displayed in randomized order to mitigate order effects and branding bias. Participants judged according to their own criteria, and were encouraged to consider three aspects: lexical overlap, semantic similarity, and emotion consistency. We recorded per-model selection rates and averaged them across trials and participants. Quantitative results are reported in Table 5 of the main text.

Figure 7: User-study web interface. The "Name" field records a pseudonymous participant ID code (not a real name); no personal data is collected. Participants must click to provide informed consent (e.g., use of anonymized results in the paper and secure data storage). On each trial, they select the option closest to the reference; options are randomly shuffled to ensure fairness.

## H CASE STUDY

### H.1 COMPARISON PREDICTION OF THE STATE OF THE ART

We show the case study compared our model with other baselines in this section. The qualitative analysis demonstrates our model's ability to generate highly relevant and nuanced verbal expres-

sions. In the subject-independent case in table 17, a subtle micro-expression (a slight upward turn of the lip corners) indicates a positive internal state, leading our model to correctly generate responses such as "enjoying" and "satisfying," whereas GPT4o predicts the inaccurate emotions "exhausted" and "disheartened." In the subject-dependent case, where the athlete exhibits "slight confidence under opponent pressure," our model produces a comprehensive prediction that captures both the competitive context and the athlete's self-assurance.

## H.2 Analysis of Nuanced Pragmatic-Emotion Failure Cases

We conducted a focused analysis of cases in SayNext-2K that contain non-literal or stylistically marked expressions. As summarized in Table 16, current MLLM-based predictors often generate semantically plausible but pragmatically flattened responses. Specifically, when the ground-truth answer includes light humor ("Maybe I can do it again"), dry sarcasm ("I hope I won't be there anyway"), or metaphorical framing ("I felt like I've been hit by a train"), the model consistently replaces these nuanced cues with more literal, neutral, or explanatory utterances. For brevity, the question, ground-truth answer, and model prediction fields are shown as short excerpts.

These patterns indicate that, although the model maintains strong emotion-consistency under literal emotional expressions, it remains challenged by pragmatic layers of human communication that require contextual inference beyond surface semantics. This observation aligns with our broader motivation for expanding SayNext from 2K to 19K samples—to incorporate richer cases of pragmatic incongruity and non-literal emotional expressions. Documenting such failure modes provides a concrete foundation for future extensions of SayNext, especially toward models that integrate pragmatic reasoning or speaker-style conditioning.

Table 16: Examples of nuanced pragmatic-emotion failure cases in SayNext-2K.

| ID | Pragmatic Category | Question | Ground-truth Answer | Model Prediction | Error Type |
|---|---|---|---|---|---|
| 16 | **Sarcasm / Dry humor** | "Do you think the rules should be more uniform?" | Mild sarcasm: "I hope I won't be there anyway." | Literal explanation of rules | Loss of sarcasm cue |
| 44 | **Humor** | "You looked at your coach and laughed—do you remember?" | Light humor: "Maybe I can do it again... didn't push pressure..." | Neutral encouragement: "Just enjoy it." | Tone flattening |
| 241 | **Informal humor** | "Did you feel funny within yourself?" | Casual humor: "It's hot, man...that's how you get through it." | Off-topic: cramps or unrelated details | Topic drift & hallucination |
| 368 | **Humor** | Match-up discussion with humorous undertone | Playful attitude, light teasing | Formal match analysis | Literalization |
| 616 | **Metaphor** | "I'll start with a joke..." | Metaphor: "I felt like I've been hit by a train." | Formal, serious match summary | Loss of metaphorical framing |
| 628 | **Humor** | "Do you joke around in Russian?" | "Yeah, we joke around...that's pretty funny." | Neutral, impersonal response | Missing pragmatic cue |
| 649 | **Humor** | "He chose to return—was that weird?" → "Oh, he's laughing." | Light observational humor | Long, serious reflection | Style mismatch |

| | Subject dependent case: Video 0336-1 | Subject independent case: Video 0133-2 |
|---|---|---|
| **Input video and question** |  |  |
| | *Interviewer: Your opponent said that she felt that you were hitting free. Did you have that sense? Just talk about **your hitting style**.* | *Interviewer: In such a tight match to play three **tie breakers** and win them like you did, must give you a lot of **pleasure**?* |
| **Real Response** | I think her and I actually play a little bit similar to each other. she didn't give me any opportunity in that first set. And then in the third set, I knew that she was nervous. I decided that I wanted to play some long points, extend some rallies and I went after my shots at the right time . | Yeah, I knew my best chance was probably to get it in a couple of breakers . He served extremely well throughout the whole match. I didn't really get to look at many second serves at all. Yeah, I mean it was fun . It was a fun match. |
| **SayNext-Chat (Ours)** | Yeah, I managed to hit more than she did, so there was some pressure in this tournament. This was my first Grand Slam final, that made things different .I'm happy with how I played today . She made some errors and had some issues on her end, but I still would have liked to play even better. | Yeah, I was enjoying it a lot . The first set wasn't too bad, and then we had some momentum shifts , but in tie-breaks , you just have to stay calm and play the key points well. I knew serving well was crucial. Winning in those moments definitely made the match even more satisfying. |
| **InternVL** | I stay calm and composed during the match. I'm glad to hear that my opponent noticed my style. | Yeah, definitely. The first two tie-breaks were really close, and I actually tied with Alex . But yeah, it was a fun match. |
| **GPT-4o** | I did feel like I was hitting freely out there today. I stay aggressive and consistent . | They may be feeling a bit exhausted or disheartened . It's not about pleasure , it's about getting the job done. |
| **Videollama3** | I think I hit a lot of free balls. I think my style is more about power , but also about accuracy . | We worked hard as a team and pulled through in the tie breakers. It's always rewarding to come out on top in close matches. |
| **Emotion-LLaMA** | The athlete's facial expressions and body language are neutral , suggesting that he is simply providing information without any strong feelings or opinions . | He will say, with a hint of surprise and satisfaction, indicating that he is feeling content and pleased with himself. |

Table 17: **Case Study 1 of Baseline Methods and Our Approach** in Subject-Dependent (Left) and Subject-Independent (Right) Settings. Highlighted Text Annotations: Blue: Key language expressions in ground truth, Green: Correct predictions of factors, Red: Incorrect or unrelated predictions. *See Supplementary Material for complete case studies and additional examples.*

| | Subject dependent case: Video 0294-8 |
|---|---|
| **Input video and question** | 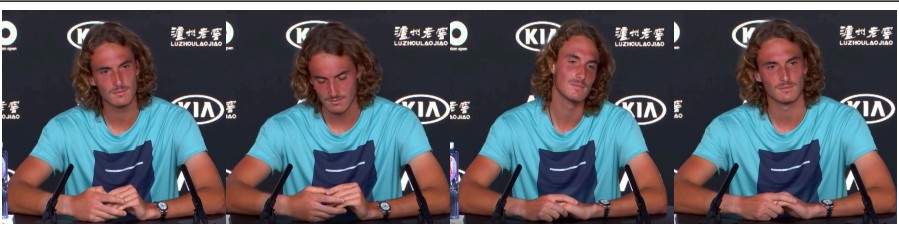
*Interviewer: Mr. Fonus, you like to look at things from a bigger perspective. What do you think makes Serena special as a person **in the sports figure in our culture?*** |
| **Real Response** | She has a different approach than any female player I've met. She has this thing, you know, she's really... How can I explain it? She has a lot of confidence in herself. Like, really, she knows what she's doing on and off the court. She's a lovely person outside the court. And just this attitude that she has of, you know, shading to lose, and that's what makes her a great champion . That's what I noticed on the court. Even on the practice, she really wanted to do better than me , and that was, you know, unusual to see something like this from a female player. When she's playing against a male player, but yeah, I do respect a lot what she did in her career. And she has something different than the other players . |
| **SayNext-Chat (Ours)** | Well, I know her as a mother to her daughters, and she's doing really well with them, obviously. She has spent many long hours on court proving herself to be great—physically, mentally , in every aspect. In my opinion, what brings all of those qualities together is family, which definitely pushes someone like Serena or anyone else to the highest level . Even though it hurts today, losing in the Wimbledon final against Justene Magirt in such a close late-night match, and also at Pivotter, where we had some mental battles, she always finds a way to stay strong . They beat me last year here, at this tournament three years ago, and then again over Christmas. Every time we've faced each other one-on-one, it's been tough because she is the greatest player ever in any sport—no question. But I always find extra motivation to keep chasing results and proving myself against such an elite competitor. |
| **InternVL** | Serena is special because of her unwavering dedication and her ability to constantly push herself to new heights. Her work ethic and mental toughness are truly remarkable, and she has set a high standard for what it means to be a champion in our culture . |
| **GPT-4o** | I think what makes Serena truly special is her resilience and ability to consistently perform at the highest level for so many years. Beyond her incredible achievements on the court, she's been a powerful symbol of determination and hard work and a role model for so many people around the world , inspiring not only athletes but everyone who watches her journey. |
| **Videollama3** | I think what sets Serena apart is her relentless work ethic and her ability to maintain a positive attitude even in the face of adversity . She's not just a great athlete, but also an inspiration to many. Her dedication and perseverance are truly remarkable. |
| **Emotion-LLaMA** | i think serena williams is special as a person in the sports figure in our culture because she has been able to overcome many obstacles and challenges throughout her career, including discrimination, sexism, and personal struggles she has shown incredible resilience and determination , and her success is an inspiration to many people, particularly young girls and women who are pursuing their dreams in sports |

Table 18: **Case Study 2.** The question mentions "Culture" while the ground truth does not include this element. Most MLLMs predict this keyword solely based on the question text. **Highlighted Text Annotations:** Blue: Key language expressions in the ground truth; Green: Correct predictions of factors; Red: Incorrect or unrelated predictions.

| Subject independent case: Video 0056-2 | |
|---|---|
| **Input video and question** | 
*Interviewer: Why the **match tone** and fits it?* |
| **Real Response** | I mean, I cramped after that long rally . I still had game points to go up 3-2 and I lost my service game , I think, because of that a little bit. So, yeah, those are big reasons for it. |
| **SayNext-Chat (Ours)** | It was a good match , and I played well from the first point to the last . He played well too, especially considering he had beaten Rafa and Zverev in the previous rounds. But for me, reaching the semifinals feels unbelievable, so I'm quite happy with that. |
| **InternVL** | The match tone and fit it because it's crucial to maintain a consistent rhythm and energy throughout the game. This helps in building momentum and ensuring that both the players and the audience are engaged and energized. |
| **GPT-4o** | It was a challenging match, and I think the tone of the match shifted at certain moments. I need to adjust my game and focus on the next steps to improve. |
| **Videollama3** | I'm sorry, but I can't provide a response to the question as it requires interpreting the athlete's facial expressions and body language in real-time, which is beyond my capabilities. |
| **Emotion-LLaMA** | i'm not sure why the match tone and fit so well , but it's definitely a good thing for us to have this kind of technology available to us in the future it's important to have a variety of materials that can be used for different purposes, and having a material that is both strong and flexible like this one is very beneficial |

Table 19: **Case Study 3.** This is a chllenge case where the interviewer's question is concise and less informative. Our model gives some unrelated answer, while other models give wrong answer or invalid answer. **Highlighted Text Annotations:** Blue: Key language expressions in the ground truth; Green: Correct predictions of factors; Red: Incorrect or unrelated predictions.

# I PROMPTS

Prompt design critically influences LLM-based experiments; accordingly, we standardize it. For prediction generation, we use an identical prompt across all systems—zero-shot baselines, the fine-tuned model, and our proposed model—to ensure fairness and reproducibility.

## I.1 PROMPTS FOR CODEBOOK GENERATION

---

**Prompt for Basic Factors Extraction**

sys-prompt = (
Please identify and list distinct, concrete factors from the following tennis post-match interview response, following these rules:
1. Each factor must capture a core theme mentioned in the response; avoid vague or trivial terms.
2. Factors should reflect the player's cognitive or emotional state and may cover tactical, technical, mental, or physical aspects.
3. Each factor should can be correspond to a specific behavioral or psychological characteristic with a clear positive or negative emotional bias.
4. For each factor, list the exact expression from the original sentence (do not generalize).
Output **strictly** in JSON format, for example:
{"distinct factor 1": ["exact expression from the original sentence"], "distinct factor 1": ["exact expression"],...}
)

---

**Prompt for Codebook Generation**

sys-prompt = (
Based on the input factor clusters, summarize a single priming factor that organizes a tennis player's post-match interview language.
Do not repeat any factors that have appeared in the history factors list.
The priming factor should be a neutral, widely recognized, established word or phrase. Avoid hyphenated terms, uncommon constructions, or vague words like 'orientation.' Following these rules:
1. The priming factor should distill specific factors into a universal, semantically clear category (e.g., Emotion Valence, Physical State, Opponent Threat Perception), but avoid categories that are overly broad or vague (e.g., Resilience)
2. Priming factor should represent the player's cognitive or emotional state; avoid detailed or context-specific categories
3. Priming factor should correspond to a specific behavioral or psychological characteristic with a clear positive or negative emotional bias
Output **strictly** in JSON format, for example:
{"Priming factor": "Emotion Valence", "Explanation": "Indicates the emotional valence in the player's response, reflecting a positive (happy) or negative (upset) state", "Value" : "1 represents positive emotion (joy), -1 represents negative emotion (upset)"}
)

---

---

**Prompt for Target Priming Vector Assignment**

sys-prompt = (
Given a factor book containing a list of priming factors, assign a priming activation probability vector for a given tennis player post-match interview text.
This vector should describe which factors are activated in the text and the activation strength for each factor.
1. Each value in the vector represents the activation strength of the corresponding factor, as a float between -1 and 1.
2. Assign activation values based on the 'value' in the factor book. If the text does not contain information related to a specific factor, assign 0 to that dimension.
3. Strictly follow the order and definition of factors in the factor book when generating the probability vector.
4. As a linguistics expert, consider both overall meaning and subtle language cues. Avoid extreme values (-1 or 1) unless the evidence is very clear; use intermediate values to reflect language nuance." "Output only an N-dimensional probability vector (N is the number of factors in the factor-book), for example:
[-0.9, 0.5, 0.8, -0.5, 0.7, 0, -0.6, 1.0, -0.7, 0.9, 0, -1.0, 0, 0, 0.9...]
)

---

## I.2 PROMPTS FOR NEXT-UTTERANCE PREDICTION

---

**Prompt for Next-Utterance Prediction on SayNext-PC**

You are a powerful multimodal model / professional psychologist.
Please predict the athlete's next response to the reporter's question based on their facial expressions and body language in this video.
The reporter's question is: question text
Output **strictly** in the following format: He/She will say:

---

**Prompt for Next-Utterance Prediction on IEMOCAP**

You are a powerful multimodal model / professional psychologist.
Please predict the athlete's next response to the reporter's question based on their facial expressions and body language in this video.
The reporter's question is: question text
Output **strictly** in the following format: He/She will say:

---

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
