# OpenReview forum: "SAYNEXT: A Benchmark and Cognitively Inspired Framework for Next-Utterance Prediction with Multimodal LLMs"
_ICLR.cc/2026/Conference — Submitted to ICLR 2026_

### Official Review · Reviewer_AYCo · 2025-10-18

**Soundness:** 3
**Presentation:** 3
**Contribution:** 2
**Rating:** 6
**Confidence:** 3

**Summary:**

This paper introduces SAYNEXT, a new benchmark and cognitively inspired multimodal framework for next-utterance prediction, aiming to enable LLMs to anticipate human dialogue responses by leveraging both verbal and non-verbal cues such as gestures, gaze, and emotion.

**Strengths:**

1. Proposes an original task of next utterance prediction grounded in cognitive science, linking dialogue modeling with human predictive processing.

2. Builds a large scale multimodal dataset SayNext PC that includes synchronized video, audio, and text data from real interactions, filling a major gap in multimodal dialogue research.

3. Designs the SayNext Chat dual route framework with learnable cognitive priming tokens, achieving consistent improvements across lexical, semantic, and emotional metrics.

**Weaknesses:**

1. Although the paper introduces a new task of next utterance prediction, it does not convincingly demonstrate its usefulness or downstream impact on other dialogue or reasoning tasks, with only brief conceptual discussion in the introduction.

2. The evaluation on the proposed dataset lacks depth and insight, offering mainly quantitative comparisons without detailed analysis of model behavior or failure cases.

3. The cognitive analogy combining dual route processing and priming is interesting but insufficiently explored, lacking stronger ablation or interpretability evidence to support the cognitive claims.

**Questions:**

see weakness

---

> ### Author Response · Authors · 2025-11-25
> **Weaknesses (1/2)**
>
> Thank you for the thoughtful feedback. We sincerely thank the reviewer for recognizing the originality of our cognitively grounded task, the value of the large-scale multimodal SayNext-PC dataset, and the effectiveness of our SayNext-Chat framework with learnable cognitive priming tokens.
>
> Firstly, we respectfully direct you to our general response, where we clarify the motivation, novelty, and overall positioning of our work in greater detail. Next, we will address your weaknesses and questions below:
>
> **Weaknesses:**
>
> **W1: Usefulness or downstream impact**. Thank you for raising this important question. Beyond scientific curiosity, enabling LLMs to predict forthcoming utterances has a clear practical impact, which we now make more explicit. As described in our revision (Sec.5 Limitations & Future Works):
>
> *“SayNext (next-utterance prediction) is therefore designed as **a proxy task** to model aspects of human predictive processing. Performing well on this task **implicitly** includes intention inference, affective anticipation, and multimodal social reasoning—capabilities that are broadly lacking in current LLMs.”*
>
> This predictive ability has broad implications in several real-world settings (As stated in line 90 in manuscript)—for example:
>
> (1) AI alignment and safety in conversational systems.
> Anticipating the user’s next utterance provides an early signal of potential misunderstandings, emotional escalation, or harmful conversational trajectories (e.g., detecting when a user is about to commit suicide). This allows the system to intervene proactively—something traditional reactive dialogue models cannot accomplish.
>
> (2) Cognitively-aware social robotics and embodied AI.
> Unlike today's passive, reactive LLM paradigm, a model with active cognitive awareness—one that can anticipate a human’s forthcoming utterance and intention—opens the door to the next generation of intelligent and embodied interaction. Such predictive capability allows robots and embodied agents to prepare actions before the human finishes speaking, coordinate turn-taking more naturally, and adapt to subtle emotional or behavioral cues in real time. This shift from passive response to proactive anticipation represents a fundamental step toward more fluid, human-like, and socially intelligent AI systems.
>
> More broadly, next-utterance prediction brings LLMs closer to human-like predictive processing, improving trust, coordination, and robustness in interactive AI systems, which is full of scientific interest. We will integrate the above demonstrations in the paper to better reveal the value of our work. Thank you for your feedback.
>
> **W2: Model behavior or failure cases**. The insights from our experiments are summarized in Section 4.2.1 (Overall Takeaways), where we highlight several key findings, including clear improvements brought by the vision modality, the consistent advantage of SayNext-Chat over baseline MLLMs, the strong contribution of priming vectors to emotional alignment, and the robust cross-scenario generalization and scalability of SayNext-Chat. These insights not only provide an examination of frontier MLLMs’ predictive capability, which has not been discussed in prior work, but also offer analysis specific to the SayNext task and the priming factor method. In addition, we **have included a more detailed case study** (includes failure case) in the appendix, as shown in **Appendix H**.

---

> ### Author Response · Authors · 2025-11-25
> **Weaknesses (2/2)**
>
> **W3: Stronger ablation or interpretability evidence**.
>
> Thanks for your constructive feedback. We are now conducting **additional ablation studies** on another model (VideoLLaMA3, in Table 2 in our revised manuscript), which will further demonstrate how the introduced priming module improves multiple metrics, particularly emotion-related ones.
>
> We are currently working intensively to finalize these additional experiments, and we kindly ask the reviewers to allow us 1–2 days to include the results.
>
> Various ablation studies regarding priming factor design in Table 1, different modalities (w/o vision) in Table 9 and 10, and different priming factor numbers in Fig 6 jointly demonstrate the robustness and necessity of the current design of our priming mechanism, showing that the learned priming factors consistently improve performance across lexical, semantic, and emotional dimensions. These analyses further validate that the proposed priming framework is not only effective but also stable across architectural choices and modality configurations.
>
> **Interpretability evidence**. The priming factors we extract—such as Perceived Pressure, Affect, Motivation, Self-Efficacy, Mental Focus, and others—are implemented as learnable cognitive tokens, which provide a meaningful and interpretable bridge between multimodal signals and the model’s internal reasoning process. These factors can naturally serve as cognitively grounded explanatory variables, offering a window into how the model anticipates a speaker’s forthcoming utterance.
>
> Given the scope and positioning of this paper—namely benchmark construction, task formulation, and algorithmic contribution—we intentionally keep the cognitive interpretation concise to remain aligned with ICLR’s primary focus on machine learning methodology. However, we fully agree that the cognitive significance of these learnable priming tokens is an important direction. In future work, we plan to collaborate with cognitive science experts to conduct a deeper, domain-grounded analysis of how these learned priming factors relate to human cognitive states and social predictive processing.
>
>
> We hope the clarifications and additional results above effectively address your concerns. As shown, we have invested substantial effort in presenting this pioneering exploration of LLMs with ambitious goals to benefit the community, and **we would greatly appreciate it if your final evaluation could reflect this and further encourage our work**.

---

> ### Author Response · Authors · 2025-11-27
> **Additional experiment of W3**
>
> Thank you for your patience. We include additional experiments integrating our priming-based framework into another MLLM model. Given the time constraints, we were only able to implement one additional MLLM as a new backbone to further demonstrate the value and generality of our priming approach. Since **VideoLLaMA3-7B** is the strongest baseline after InternVL, we believe it's reasonable to choose it as the candidate and adapt our priming mechanism to its architecture, including modifying its input pipeline and integrating learnable cognitive tokens, and fine-tuning.
>
> The results are provided below and will be incorporated into Table 2 of our revised manuscript.
>
> Table 2. Relative deltas (higher is better) w.r.t.*VideoLLaMA3-7B*. Our proposed priming approach effectively improves all metrics.
>
> | Method               | ΔBLEU-4/% | ΔROUGE-L/% | ΔBERTScore-F1 | ΔBERT sent. cos. | ΔEmot. Val. | ΔEmot. Arou. |
> |----------------------|-----------|------------|----------------|-------------------|--------------|----------------|
> | Factor in Prompt     | -0.425    | -2.189     | -0.01173       | -0.04620          | -0.01678     | -0.02802       |
> | Vector in Prompt     | -0.445    | -1.931     | -0.00570       | -0.05310          | -0.02608     | -0.04135       |
> | Only Finetune        | +1.309    | +3.586     | +0.02332       | +0.04560          | +0.06546     | +0.04309       |
> | **Finetune+Priming (Ours)** | **+1.558** | **+3.816** | **+0.02501** | **+0.05930** | **+0.07069** | **+0.04652** |
>
> As you can see, our priming approach can be used as a plug-in that can be seamlessly **adapted to different existing MLLMs**, consistently yielding notable improvements on SayNext-Bench beyond what standard fine-tuning alone can achieve.

---

> ### Author Response · Authors · 2025-12-02
> **Rebuttal Summary of Reviewer AYCo**
>
> We sincerely thank the reviewer for recognizing the **originality of our cognitively grounded task**, the **value of the large-scale SayNext-PC dataset**, and the **effectiveness of our priming-based SayNext-Chat framework**. In the rebuttal, we provided complete responses to all identified weaknesses.
>
> The reviewer’s main concerns involved the usefulness and downstream impact, model behavior and failure cases, and stronger ablation/interpretability evidence. We addressed these by:
>
>  1. **Usefulness or downstream impact (W1)**: Clarifying the broader significance and practical impact of next-utterance anticipation and integrating these explanations into the revised manuscript.
>
>  2. **Failure cases (W2)**: Highlighting the key findings in Section 4.2.1 and adding more detailed case studies, including failure cases, to Appendix H.
>
>  3. **Stronger ablation (W3)**: Conducting **additional ablation experiments on VideoLLaMA3** (Table 2), which required substantial adaptation effort, and further validate the generality of our priming mechanism.
>
>  4. Interpretability (W3): Strengthening interpretability discussion by explaining how the learned priming factors offer cognitively meaningful explanatory signals.
>
> We also refined related explanations across the manuscript. We greatly appreciate the AC’s time, and we kindly hope the final evaluation can acknowledge the substantial effort and contributions of this work.

---

### Official Review · Reviewer_7tXM · 2025-10-30

**Soundness:** 2
**Presentation:** 3
**Contribution:** 2
**Rating:** 4
**Confidence:** 4

**Summary:**

This paper introduces a new task, namely human next-utterance prediction. The authors argue that predicting human utterances reflects LLMs' ability to better understand human emotions and other needs, which is beneficial for building LLMs applied in scenarios involving intensive human interaction, such as embodied intelligence. Therefore, this paper proposes a benchmark, SayNextBench, to evaluate LLMs' ability to understand information and predict human next utterances in multimodal scenarios. The benchmark includes SayNext-PC2K and SayNext-PC19K.

To evaluate the models, this paper proposes four complementary evaluation metrics: Subject-Dependent Evaluation, Subject-Independent Evaluation, Cross-Scenario Evaluation, and Scalability Evaluation.

Additionally, this paper proposes a dual-route prediction MLLM, SayNext-Chat, and experiments show that its ability to predict human next utterances is significantly stronger than other zero-shot models.

**Strengths:**

This paper is well-presented, rigorously organized, and well-written.

This paper attempts to propose a novel task, construct a relatively comprehensive benchmark, and train a model, making the work quite thorough.

**Weaknesses:**

I will place the weaknesses and my questions together in this section.

I may have some misunderstandings about the motivation of this paper. I do not fully understand the difference between the next-utterance prediction task and dialogue tasks, as well as the necessity of proposing this as a separate task. If an LLM can effectively model the contextual logic in a dialogue, then predicting a human's next utterance that aligns with the context should not be a problem. What is the difference between multimodal next-utterance prediction and multimodal dialogue? Does it lie in responding to human A's dialogue or predicting human A's next utterance? With all due respect, there is no difference between the two—predicting human A's next utterance is equivalent to responding to human B's dialogue. In early dialogue tasks, most datasets were constructed by collecting dialogue histories between two human annotators.

Perhaps the task proposed in this paper would be more appropriately termed fine-grained human intent classification or prediction, and the authors need to emphasize the importance of fine granularity and its relationship with previous fine-grained intent recognition work.

Additionally, there is a one-to-many issue here, meaning the provided reference responses are not unique. In this paper, the evaluation mainly involves calculating token and semantic relevance between predicted responses and reference responses, without assessing the consistency between predicted responses and the context or other multimodal contexts. Human evaluation comparing the quality of responses across models can alleviate my concerns to some extent.

In Section 2.3, the paper mentions four evaluation perspectives, which also serve as guiding principles for constructing the benchmark. Is there a correspondence with the subsequent experimental analysis? It would be better if the subsequent experimental analysis and the models' performance in these four capabilities could be aligned.

**Questions:**

N/A

---

> ### Author Response · Authors · 2025-11-25
> **Weaknesses**
>
> Thank you for the thoughtful feedback. We sincerely thank the reviewer for the encouraging comments. We truly appreciate the recognition that the paper is well-presented and rigorously organized, and that our efforts in proposing a novel task, constructing a comprehensive benchmark, and developing a dedicated model make the work thorough. Your positive feedback is highly motivating for us.
>
> Firstly, we respectfully direct the reviewer to our general response, where we clarify the motivation, novelty, and overall positioning of our work in greater detail. Next, we will address your weaknesses and questions below:
>
> **Weaknesses**:
>
> **W1: Motivation**.
>
> We kindly encourage you to refer to our general response, where we comprehensively illustrate our motivation, significance, and purpose of this work. To conclude, our task aims to mimic human cognitively-involved communicative modeling (actively using multi-modal verbal and non-verbal cues to achieve predictive coding), instead of a passively predicting next token task, in which a linguistically coherent and high-likelihood reply (seen in current LLMs) is NOT enough for our proposed SayNext task.
>
> **W2: Fine-grained human intent classification**.
>
> We address your concerns in the general response, regarding the position of our paper. As you suggested, in the revision, we will explicitly clarify SayNext’s relationship to prior work on fine-grained human intent classification/prediction as follows. Thank you for your constructive feedback.
>
> We added:
>
> *“SayNext (next-utterance prediction) is therefore designed as **a proxy task** to model aspects of human predictive processing. Performing well on this task **implicitly** includes intention inference, affective anticipation, and multimodal social reasoning—capabilities that are broadly lacking in current LLMs. Consequently, we remind readers that the SayNext task does not fit neatly into existing categories such as emotion recognition, intention classification, or dialogue continuation. Instead, it opens a distinct and complementary research direction that we hope will motivate future work in cognitively grounded predictive modeling and inspire next-generation LLMs with more human-like anticipatory abilities.”*
>
>
>
> **W3: One-to-many issue**. Thank you for pointing out the one-to-many nature of open-ended generation. We fully agree that a single reference response is not unique, which is precisely why traditional text-matching metrics are insufficient for this task. To address this, our evaluation does not rely solely on reference overlap; instead, we introduce metrics that explicitly assess contextual consistency, semantic alignment, emotion appropriateness, and multimodal coherence, along with human/LLM preference judgments. These metrics evaluate whether the predicted response is plausible and contextually valid, not whether it matches the reference verbatim.
>
> Therefore, while the task is inherently one-to-many, our evaluation framework is specifically designed to capture the many-to-many nature of real conversational anticipation.
>
> **W4: Performance and evaluation perspectives**. Thank you for this helpful suggestion. The four evaluation perspectives introduced in Section 2.3 indeed serve as the conceptual foundation of the benchmark, and they correspond directly to the four groups of metrics in our experiments. We will make this correspondence more explicit in the revision by clearly annotating, after the four protocols described in Section 2.3, the evaluation unit associated with each protocol. This clarification will help readers better understand how the experimental results reflect the model’s performance across the four core capabilities.
>
> We hope the clarifications and additional results above effectively address your concerns. As shown, we have invested substantial effort in presenting this pioneering exploration of LLMs with ambitious goals to benefit the community, and **we would greatly appreciate it if your final evaluation could reflect this and further encourage our work.**

---

> ### Author Response · Authors · 2025-12-02
> **Rebuttal Summary of Reviewer 7tXM**
>
> We sincerely thank the reviewer for the positive and encouraging comments on the **clarity, organization, and thoroughness of our work**. In the rebuttal, we provided complete responses to all identified weaknesses.
>
> The reviewer’s main concerns involved the task motivation, difference from fine-grained human intent prediction, one-to-many generation, and clarity of evaluation perspectives. We addressed these by:
>
>  1. **Motivation (W1)**: Providing a detailed clarification of the task motivation and positioning in the general response and integrating this into the revised manuscript.
>
>  2. **Position (W2)**: Explicitly clarifying SayNext’s **difference from fine-grained intent prediction** and highlighting its role as a **cognitively grounded proxy task** for anticipatory modeling.
>
>  3. **One-to-many issue (W3)**: Explaining how our evaluation framework is designed to handle the inherent one-to-many nature of open-ended generation through semantic, emotional, multimodal, and preference-based measures.
>
> We also refined relevant explanations throughout the manuscript. We greatly appreciate the AC’s time, and we kindly hope the final evaluation can acknowledge the substantial effort and contributions of this work.

---

### Official Review · Reviewer_WYoH · 2025-11-03

**Soundness:** 3
**Presentation:** 4
**Contribution:** 2
**Rating:** 6
**Confidence:** 4

**Summary:**

This paper proposes a benchmark for predicting the next utterance in a dialogue in the context of a sports interview, where there is access to the previous dialogue turn (the interviewer's question) and video from the only the interviewee (the sports star) whilst this question is being asked.

The model is used to generate natural language responses, which are compared to the ground truth response using a variety of metrics.

The paper also proposes, and experiments with, a novel cognitively-inspired "priming" technique, in which a code book of "priming factors" are created.  At inference time, numerical values for these factors can be predicted, where are used in turn to better inform the response generation task.

The dataset, SayNext-Bench is used to train a model, which is compared to zero-shot generation using multi-modal LLMs.

**Strengths:**

I found this to be an enjoyable and interesting paper.  It is not straightforward to collect this kind of multimodal dialogue dataset, and it is clear that a great deal of work has gone in to it.  Although very constrained in terms of its domain, I'm sure it will be a useful resource, particularly for researchers wishing to study emotional dialogue – on account of the domain, the corpus is unusually rich in emotional dialogue.

Although I didn't find all of the metrics investigated very convincing, and the setup does have some limitations (see below) the emotion metrics will be useful, and the corpus representations a great multimodal emotion prediction + generation task.

The work was clearly motivated and had good use of examples and diagrams.

**Weaknesses:**

I wasn't entirely clear what the primary purpose of the paper was, making evaluation somewhat difficult.  As a benchmark, it is good, for the reasons mentioned above.  However, one limitation is the narrowness of the domain, being constrained not just to sports interviews, but in fact to tennis interviews.  Another limitation is that only the single interviewer turn is included, meaning that it is impossible to generate accurate responses that rely on past dialogue context, or in fact, to events that had happened in the match immediately preceding that would be common ground to both speakers (this is clear from the ground truth examples, which often refer to this information).

The authors take care to use metrics that circumvent these limitations, but it does mean that metrics such as BLEU and ROUGE – and to some extend, the BERT metrics – are effectively meaningless, since no generation can achieve anything close to the ground truth on these scores.  (The authors do acknowledge these issues).

I suspect that it also means that the model fine-tuned on the benchmark's training set have an unfair advantage over the zero-shot LLMs, which are not as far as I can tell, provided with any tennis-related context.  This makes the comparisons less useful.  I'm not sure why you didn't consider fine-tuning other multimodal models, particularly so that you could have more convincingly demonstrated the value of the priming approach (which was of the most interesting parts of the paper).

Another limitation is the lack of video for the interview, meaning that the benchmark will not generalise to many more natural human-human dialogue settings.

I suggest that at its core, this is not really a dialogue prediction task, but rather an video emotion-prediction task, with emotion predictions made (rather cleverly) through the medium of a text-based response.  This is valuable in itself, but probably should be more clearly acknowledged.

If on the contrary, it really is a text prediction task you are aiming at, there should have been more comparison to text-only models in the main paper (there seemed to be just a limited number of these comparisons in the appendix), much more context added, and metrics should have included figures such as perplexity, which would have avoided the major problem of responses all having low lexical overlap.  You should also have considered citing the wealth of related turn prediction task that already exist in speech and text based dialogue.

**Questions:**

I struggled to understand the way in which IEMOCAP is adapted to SayNext-Bench (§4.2.4) – I really could not understand how this was done, and also why the figures for LO in the bottom row of Table 2 are so dramatically better than all the others.

---

> ### Author Response · Authors · 2025-11-25
> **Weaknesses (1/2)**
>
> Thank you for the thoughtful feedback. We sincerely appreciate the reviewer’s thoughtful and encouraging comments, especially regarding the difficulty of collecting such multimodal dialogue data, the emotional richness and usefulness of the corpus, and the clarity and motivation of the work.
>
> Firstly, we respectfully direct the reviewer to our general response, where we clarify the motivation, novelty, and overall positioning of our work in greater detail. Next, we will address your weaknesses and questions below:
>
> **W1: Primary purpose**. We kindly ask you to refer to the general response, where we comprehensively illustrate our motivation and purpose of this work.
>
> **W2: Multi-turn/single-turn**. We fully agree with the reviewer. As noted in Section 5 (Limitations & Future Works), multi-turn dialogue could indeed help capture speaker-specific habits and discourse dynamics. However, extending SayNext to a multi-turn setting involves substantial additional design choices — such as how to segment long interaction histories, how to adapt the MLLM model architecture to support explicit memory mechanisms, and how to redesign the evaluation metrics to account for context accumulation and temporal dependencies. This is beyond the workload that can be contained in one conference paper. Thus, we focus on the one-turn setting in this first version, where the main focus is to present the SayNext benchmark that already contains a substantial workload, as you can see. Nevertheless, multi-turn prediction is a natural extension of SayNext, and we already plan it as an important direction in the next step of our future work upon the acceptance of the SayNext benchmark.
>
> **W3. BLEU and ROUGE metrics**. Thank you for pointing this out. We fully agree that metrics such as BLEU/ROUGE (and partly BERT-based scores) cannot meaningfully reflect performance in open-ended generation tasks like ours. This is precisely why we also introduced metrics specifically designed to overcome these limitations (emotion, semantic consistency, preference-based human manual/MLLM judgments), which we believe provide a much more faithful assessment of the task. However, traditional metrics like BLEU are still widely used in the NLP community, so we report them only as defensive baseline checks to highlight the inadequacy of conventional NLG metrics in this setting, rather than as indicators of final model quality.
>
> **W4. Fine-tuned multimodal LLMs**. Thank you for this insightful observation. We agree that fine-tuning on the benchmark provides task-specific grounding that zero-shot multimodal LLMs do not possess. However, this distinction is intentional and directly aligned with the core motivation of our work.
>
> As shown in our preliminary experiments, even state-of-the-art conversational and multimodal LLMs—GPT-4o, Gemini 2.5, VideoLLaMA3, InternVL2, LLaVA-NeXT, and InstructBLIP—struggle to produce semantically appropriate next-utterance predictions despite being provided with explicit task instructions and visual context. This fundamental limitation persists regardless of whether the domain is tennis or any other topic.
>
> Our fine-tuned model is therefore NOT intended as a “fair competitor” to zero-shot LLMs, but as a proof-of-concept demonstrating:
>
> 1) The inherent limitation of current MLLMs—rooted in passive next-token prediction—when faced with human-intuitive tasks that require active cognitive anticipation, and
>
> 2) How our cognitively inspired priming mechanism, when combined with fine-tuned MLLMs, can more effectively address this challenge.
>
> Importantly, as shown in our paper, we also explicitly conduct the fair comparison experiment between our method with finetuned MLLMs.  Table 1 in our original paper already shows that compared to a purely fine-tuned model (InternVL), our cognitively inspired SayNext-Chat further improves performance across metrics, reinforcing the value of the proposed priming mechanism.
>
> To further address your concern, we include **additional experiments** integrating our priming-based framework into other MLLM models. The results will be provided below, and Table 2 in our revised manuscript.
>
> We are currently working intensively to finalize these additional experiments, and we kindly ask the reviewers to allow us 1–2 days to include the results.

---

> ### Author Response · Authors · 2025-11-25
> **Weaknesses (2/2)**
>
> **W5: Generalization issue of interview videos**. Thank you for highlighting this limitation. We agree that our current setting includes only the interviewee’s video, which differs from fully bidirectional human–human interaction. However, **this design choice is intentional** and grounded in both the task formulation and data feasibility.
>
> First, our primary goal is to model how a speaker’s nonverbal cues reveal their forthcoming utterance, which inherently focuses on the interviewee rather than the interviewer. Because we need the camera to consistently focus on the subject, capturing the non-verbal behaviors to conduct the prediction of the next utterance with which a sports interview is a good choice.
>
> Second, collecting high-quality dual-view video at scale is logistically difficult and almost impossible in real-world media environments (we kindly encourage you to refer to **Appendix B.1**, where we illustrate the data feasibility issue). We kindly remind you that there is also an inherent trade-off in the data construction: natural conversational footage is unconstrained and allows free expression, but lacks controlled (multi-)camera setups, while controlled multi-view recordings sacrifice authenticity and spontaneity. Our use of sports interviews strikes a practical balance between naturalness, freedom of expression, and feasibility.
>
> Importantly, the proposed priming mechanism and benchmark protocol are not restricted to single-view input. They can be directly extended to dual-view or multi-party conversation datasets when such data becomes available. We consider our benchmark as a first step toward this broader direction.
>
> **W6: Paper position**. We kindly ask you to refer to the general response, where we comprehensively illustrate our motivation and purpose of this work.
>
> **W7: Text prediction task and more context**. We believe the general response and the response in W2 shall address your concerns.

---

> ### Author Response · Authors · 2025-11-25
> **Questions**
>
> **Q1: Adapting IEMOCAP to the SayNext**. We adapt IEMOCAP to the SayNext benchmark to evaluate cross-scene generalization, since SayNext-PC contains only sports-interview scenarios. For fairness, we apply exactly the same processing pipeline as in SayNext-19K—including constructing the priming codebook, generating priming factors, and training the model—so that performance differences reflect dataset properties rather than methodological inconsistencies.
>
> The high performance on IEMOCAP primarily arises from the intrinsic characteristics of the dataset. IEMOCAP contains much shorter and simpler utterances (avg. 4.5s) compared to SayNext-PC (avg. 22.44s for PC2K and 30.368s for PC19K), making it an inherently easier prediction task and naturally yielding higher lexical-overlap scores.
>
> To further address your concern, we additionally conduct cross-dataset validation under multiple training–testing combinations (**Appendix C.3, Table 13**), including: train on 2K → test on IEMOCAP / 19K, train on IEMOCAP → test on 2K / 19K, train on 19K → test on 2K / IEMOCAP. Across all settings, we observe that larger training sets lead to more robust generalization, confirming that the performance trends are consistent with dataset size and difficulty rather than any adaptation artifact.
>
> We hope the clarifications and additional results above effectively address your concerns. As shown, we have invested substantial effort in presenting this pioneering exploration of LLMs with ambitious goals to benefit the community, and **we would greatly appreciate it if your final evaluation could reflect this and further encourage our work**.

---

> ### Author Response · Authors · 2025-11-27
> **Additional experiment of W4**
>
> Thank you for your patience. We include additional experiments integrating our priming-based framework into another MLLM model. Given the time constraints, we were only able to implement one additional MLLM as a new backbone to further demonstrate the value and generality of our priming approach. Since **VideoLLaMA3-7B** is the strongest baseline after InternVL, we believe it's reasonable to choose it as the candidate and adapt our priming mechanism to its architecture, including modifying its input pipeline and integrating learnable cognitive tokens, and fine-tuning.
>
> The results are provided below and will be incorporated into Table 2 of our revised manuscript.
>
> Table 2. Relative deltas (higher is better) w.r.t.*VideoLLaMA3-7B*. Our proposed priming approach effectively improves all metrics.
>
> | Method               | ΔBLEU-4/% | ΔROUGE-L/% | ΔBERTScore-F1 | ΔBERT sent. cos. | ΔEmot. Val. | ΔEmot. Arou. |
> |----------------------|-----------|------------|----------------|-------------------|--------------|----------------|
> | Factor in Prompt     | -0.425    | -2.189     | -0.01173       | -0.04620          | -0.01678     | -0.02802       |
> | Vector in Prompt     | -0.445    | -1.931     | -0.00570       | -0.05310          | -0.02608     | -0.04135       |
> | Only Finetune        | +1.309    | +3.586     | +0.02332       | +0.04560          | +0.06546     | +0.04309       |
> | **Finetune+Priming (Ours)** | **+1.558** | **+3.816** | **+0.02501** | **+0.05930** | **+0.07069** | **+0.04652** |
>
> As you can see, our priming approach can be used as a plug-in that can be seamlessly **adapted to different existing MLLMs**, consistently yielding notable improvements on SayNext-Bench beyond what standard fine-tuning alone can achieve.

---

> ### Author Response · Authors · 2025-12-02
> **Rebuttal Summary of Reviewer WYoH**
>
> We sincerely thank the reviewer for recognising the **originality of our cognitively grounded task**, the **emotional richness and usefulness of the corpus**, the **difficulty of data collection**, and the **thoughtful design of our priming factors and framework**. In the rebuttal, we provided complete responses to all identified weaknesses and questions.
>
> The reviewer’s main concerns involved the task purpose, multi-turn setting, limitations of BLEU/ROUGE, and fine-tuned vs. zero-shot MLLMs. We addressed these by:
>
> 1. **Primary purpose (W1)**: Providing a detailed clarification of the task positioning in the general response and integrating this clarification into the revised manuscript.
>
> 2. **Multi-turn/single-turn (W2)**: Explaining the practical and methodological reasons for the one-turn setting and outlining multi-turn extensions as planned future work.
>
> 3. **BLEU and ROUGE metrics (W3)**: Clarifying the role of traditional metrics and highlighting more meaningful semantic, emotional, and human-evaluation measures.
>
> 4. **Fine-tuned multimodal LLMs (W4)**: Introducing a **substantial new experiment** adapting our priming framework to **VideoLLaMA3**. The results (now in Table 2 of the revised manuscript) further confirm the generality and effectiveness of our approach.
>
> We also included **new empirical results on cross-dataset validation** and refined explanations throughout the manuscript. We greatly appreciate the AC’s time, and we kindly hope the final evaluation can acknowledge the substantial effort and contributions of this work.

---

### Official Review · Reviewer_gkQJ · 2025-11-12

**Soundness:** 2
**Presentation:** 3
**Contribution:** 2
**Rating:** 4
**Confidence:** 3

**Summary:**

This study addresses a key limitation of large language models (LLMs): their struggle to accurately predict humans’ next conversational utterances, unlike humans who use multimodal cues (gestures, gaze, tone). The team developed SayNextBench (a benchmark) and SayNext-PC (a dataset of real-world dialogues like post-match interviews), plus a cognitively inspired dual-route model (SayNext-Chat). By integrating verbal and non-verbal signals and using “priming factors” to capture intent, the model outperforms state-of-the-art (SOTA) MLLMs in lexical overlap, semantic similarity, and emotion consistency, paving the way for more human-like AI dialogue.

**Strengths:**

The SayNext-PC dataset scales from 2K to 19K samples, covering diverse cultures and scenarios, while its four evaluation protocols (plus user studies) add rigor.

**Weaknesses:**

Overall, this type of data construction lacks novelty for me, and there are too many such works in the literature. Additionally, the method used is too simplistic and lacks any theoretical support.

Specifically, first, all models show low lexical overlap (max ~5%), highlighting an inherent challenge: even optimized models struggle to replicate human phrasing, revealing limitations in capturing personalized language styles. Second, the core dataset focuses on sports post-match interviews, a relatively narrow scenario. While cross-scenario validation was done, generalization to casual chats or professional settings (e.g., workplace communication) remains unproven.

Furthermore, the model relies on GPT-4.1 to extract priming factors and assign vectors, reducing openness and reproducibility—ordinary researchers may face barriers to low-cost reuse. Additionally, the study only focuses on single-turn dialogue prediction. Real conversations are multi-turn, so the model lacks long-term tracking of context coherence or speaker habits, limiting practical use.

Finally, the 20 fixed priming factors were chosen empirically; no testing was done on dynamic adjustment for different scenarios (e.g., casual vs. formal dialogue may need different cognitive-emotional dimensions). Training also requires A100 GPUs, creating high hardware barriers that hinder widespread adoption.

**Questions:**

1. Since low lexical overlap is a common issue, would adding multi-turn dialogue data or training the model to learn individual speakers’ language habits help it better mimic human phrasing?

2. The 20 priming factors were set based on experience, could this number be dynamically adjusted for different scenarios? For example, casual chats and formal interviews may require different cognitive-emotional dimensions, so would flexible factor counts improve performance?

3. Among non-verbal cues (gestures, facial expressions, tone), which contributes most to next-utterance prediction? If only core cues are retained, can we reduce computational costs without sacrificing performance?

4. Beyond post-match interviews, would the model’s performance drop significantly in more casual (e.g., friend chats) or professional (e.g., client communication) scenarios? What strategies could enhance cross-scenario robustness?

5. Currently, GPT-4.1 is used to generate priming factors, would switching to open-source models (e.g., Llama series) drastically reduce effectiveness? Are there low-cost alternatives to make this framework accessible to more researchers?

6. While emotion consistency is strong, real dialogues often include complex emotions like humor or sarcasm. Can the model accurately capture and predict such nuanced next utterances?

---

> ### Author Response · Authors · 2025-11-25
> **Weaknesses (1/2)**
>
> Thank you for the thoughtful feedback. We sincerely thank the reviewer for recognizing the scalability, cultural diversity, and rigorous evaluation design (including user studies) of the SayNext-Bench.
>
> Firstly, we kindly remind you to read the general response to better understand the motivation, novelty, and position of our work. Next, we will address your weaknesses and questions below:
>
> **W1: Dataset Novelty**
>
> The construction of such a dataset sounds intuitive, but surprisingly, there is no existing large-scale dataset readily available for our task to train and evaluate MLLMs. Indeed, there are many seemingly identical works in the literature, but most of the existing large-scale datasets originate from movies or TV series, in which the scene switch is swift and frequent, and the conversation jumps without continuity and consistency. This is not suitable for our multimodal next-utterance prediction task, where a stable, continuous view of one interlocutor is needed for better inferring communicative intent and emotion with clean conversation segmentation. Our dataset fills this gap for the community.
>
> **W2: Simplistic method**
>
> We respectfully disagree that our method is “too simplistic” or lacks theoretical grounding. The core design of our approach may sound intuitive, but it is not simple. The SayNext-Chat is explicitly motivated by predictive-coding theory and social cognition research, which emphasize that humans anticipate others’ utterances using multimodal perceptual cues rather than passive textual continuation. However, *naively* implementing our idea of priming factors via prompting tuning (fixed priming factors) will not improve the performance, and even lead to performance degradation (please see the relative deltas in Table 1 and 2). Thus, we encode those priming factors as learnable tokens to dynamically learn the beliefs and priors. To our knowledge, there is the FIRST attempt made in the field trying to integrate carefully designed (orthogonal) learnable tokens to encode and represent multi-modal cues as *beliefs* (priming factors) to achieve the SayNext task. Our formulation, therefore, is not a heuristic simplification but a theoretical paradigm change to tackle the flaws of the current LLMs' limitation. The complexity can also be partially reflected by the resource demand (A100 GPUs). Besides, SayNext-Chat is only one fold of the contributions where the benchmark is also vital to the community.
>
> **W3: All models show low lexical overlap, revealing limitations in capturing personalized language styles.**
>
> Indeed, all the models show low LO. But, this is EXACTLY the core motivation we propose for the SayNext-Bench. As discussed in the paper, line 52, this surprising shortcoming is not a trivial flaw; rather, it reflects a fundamental limitation of LLMs (please see the general response). We further argue that this unexpectedly low LO performance is not a weakness of our method, but rather clear evidence that we have identified a promising and genuinely challenging new task—one that remains difficult even in an era where LLMs demonstrate strong overall capabilities. We believe this is not a limitation of our work but rather a strong motivation and insight into what needs to be shared with the community.
>
> **W4: Cross-scenario validation**
>
> Our cross-scenario evaluation (Sec. 4.2.4, Table 2) already shows that the model generalizes reasonably well to IEMOCAP, which contains casual, daily dialogue settings. We further supplement the IEMOCAP dialogue scenarios in **Appendix B.2, Table 8, Page 16** to address the reviewer’s concern. As discussed in Appendix B.1, most existing datasets do not meet the essential criteria of the SayNext task. This gap motivated the construction of SayNext-Bench, which unifies diverse validation scenarios under a single benchmark.
>
> **W5: GPT-4.1 for priming factors**
>
> We thank the reviewer for pointing out the reproducibility cost concern. Firstly, to extract the priming factors as reported in our experimental results using GPT-4.1, the cost is only around 13 euros; we believe this is affordable for ordinary researchers. Besides, we have conducted extra experiments using Llama-3.1 (open-source models, free) and Gemini-2.5 flash (even cheaper choice) to generate priming vectors (see **Appendix F.3, Table 15**), and their performance shows no significant drop compared with GPT-4.1. Thirdly, we release all code and datasets to ensure that the entire framework can be fully reproduced by the community. Finally, while we agree that current MLLM research in the community requires substantial computational and financial resources, which is an acknowledged barrier for broader accessibility [1], this challenge is general to the AI field and unfortunately falls beyond the scope of what our work can directly address.
>
> [1] Julian Togelius and Georgios N. Yannakakis. Choose Your Weapon: Survival Strategies for Depressed AI Academics, arXiv preprint, arXiv:2304.06035.

---

> ### Author Response · Authors · 2025-11-25
> **Weaknesses (2/2)**
>
> **W6: Multi-turn tracking**
>
> We fully agree with the reviewer. As noted in Section 5 (Limitations & Future Works), multi-turn dialogue could indeed help capture speaker-specific habits and background knowledge. However, extending SayNext to a multi-turn setting involves substantial additional design choices — such as how to segment long interaction histories, how to adapt the MLLM model architecture to support explicit memory mechanisms, and how to redesign the evaluation metrics to account for context accumulation and temporal dependencies. This is beyond the workload that can be contained in one conference paper. Thus, we focus on the one-turn setting in this first version, where the main focus is to present the SayNext benchmark that already contains a substantial workload, as you can see. Nevertheless, multi-turn prediction is a natural extension of SayNext, and we already plan it as an important direction in the next step of our future work upon the acceptance of the SayNext benchmark.
>
>
> **W7: Priming factors chosen**
>
> Thank you for the question. As detailed in **Appendix F.1**, the 20 priming factors were not selected empirically or based on scenario assumptions. Instead, we derived them by evaluating the Silhouette Coefficient (SC) of multiple word-vector clusterings. Thus, the codebook is built with dimensions largely orthogonal to each other. Besides, we encourage you to look into the coding book generated, the priming factors extracted (e.g., affect, self-evaluation, physical state, etc.) can effectively adapt to various scenarios, which we believe that causal and formal chats can be covered. Thus, the priming factors arise from a data-driven and mathematically grounded selection process rather than manual scenario-dependent heuristics; the cross-scenario evaluation (Table 3) can also support this claim.
>
> **W8: Training also requires A100 GPUs**
>
> Firstly, as discussed in W5, while we agree that current MLLM research requires substantial computational and financial resources—an acknowledged barrier to broader accessibility—this challenge is inherent to the entire field rather than specific to our method.
> Secondly, the use of a ~7B-scale model is not an arbitrary decision but reflects a well-recognized consensus in the community: 7B has become the minimal capacity at which LLMs and MLLMs can reliably demonstrate meaningful reasoning, alignment, and multimodal understanding. Models below this scale (e.g., 1–2B) are widely known to suffer from severe performance degradation, unstable multimodal grounding, and unreliable conversational behavior, making them unsuitable for evaluating nuanced tasks such as multimodal next-utterance prediction.
> Finally, running a 7B model in practice requires at least a single A100-class GPU, which has already become the de facto standard hardware baseline in recent MLLM research. Our computational setting therefore follows the commonly accepted minimum requirement for conducting credible experiments in this domain, rather than reflecting an unnecessarily large design choice.

---

> ### Author Response · Authors · 2025-11-25
> **Questions**
>
> **Q1: Low lexical overlap**. please see W3 and W6.
>
> **Q2: Priming factors chosen**. Please see W7.
>
> **Q3: Various non-verbal cues**. Technically, we input whole RGB data, without differentiating different non-verbal cues, such as body gestures/ facial expressions etc. As mentioned, the core contribution is the paradigm-changing benchmark of SayNext. In the future work, it would be an interesting direction to investigate the contribution of those detailed cues, which also proves the value of our dataset.
>
> As shown in Table 9 and Table 10, the performances of the models without vision will substantially decrease, which also proves the significance of our proposed SayNext-Bench: mimicking human phrasing and cognitive modeling via next-unterance prediction is not equivalent to traditional NLP next-token prediction task.
>
> **Q4. Cross-scenes**. Please see W4. In addition, incorporating more varied conversational styles in future training data and design models with context-learning ability are straightforward strategies for enhancing robustness across scenarios.
>
> **Q5. Switch GPT4.1 to open-source LLMs**. Please see W5 and W8.
>
> **Q6. Complex emotions like humor or sarcasm**.
>
> Thank you for the insightful question. One motivation for expanding SayNext from 2K to 19K samples was to include a broader range of nuanced pragmatic emotions. Interestingly, many of our failure cases arise from non-literal expressions such as humor, sarcastic remarks, or metaphorical phrasing. Inspired by the reviewer’s suggestion, we have now categorized some nuanced-emotion failure cases and included a detailed analysis in **Appendix H.2**.
>
> We agree that this is an interesting and valuable direction. Throughout our additional case study, we show that our proposed SayNext benchmark provides a concrete foundation and valuable materials for upcoming researchers to build upon.
>
> However, we also kindly remind reviewers that sarcasm and humor constitute broad and complex research topics, which are not our research focus. This shall not be the weakness of our work, as they cannot be fully addressed within the scope of this conference paper, but rather a potential future direction of our work. We have added this point to the revised Section 5 (Limitations & Future Works) as below:
>
> *"Analyzing SayNext by investigating more nuanced pragmatic–emotional expressions, such as sarcasm, humor, and metaphor, can be a promising direction, which remains challenging even for advanced LLMs."*
>
>
> We hope the clarifications and additional results above effectively address your concerns. As shown, we have invested substantial effort in presenting this pioneering exploration of LLMs with ambitious goals to benefit the community, and **we would greatly appreciate it if your final evaluation could reflect this and further encourage our work**.

---

> ### Author Response · Authors · 2025-12-02
> **Rebuttal Summary of Reviewer gkQJ**
>
> We sincerely thank the reviewer for **the positive recognition of our dataset scalability**, **cultural diversity**, and **rigorous evaluation setup**. In the rebuttal, we provided complete responses to all eight weaknesses and six questions.
>
> The reviewer’s main concerns were:
>
> 1. **Dataset novelty (W1)**: We clarified the absence of any existing large-scale dataset for this task.
>
> 2. **Model complexity (W2)**: We highlighted our first-of-its-kind integration of orthogonal learnable priming tokens.
>
> 3. **Resource cost (W5, W8)**: we **added experiments with Llama-3.1 and Gemini-2.5**, and contextualised the broader computational demands of current MLLM research.
>
> We also supplied targeted revisions for all remaining points, including cross-scenario validation, priming-factor design, and nuanced pragmatic-emotion cases etc.
>
> We greatly appreciate the AC’s time. This work reflects substantial additional effort during the rebuttal, and we kindly hope the final evaluation can acknowledge the ambition and contribution of this submission.

---

### Author Response · Authors · 2025-11-25
**General Response**

We sincerely thank the reviewers for their thoughtful and encouraging feedback. We deeply appreciate the recognition of our work’s originality, the conceptual motivation behind our cognitively grounded task, the scalability and collecting difficulty of the SayNext-PC dataset. We are also grateful for the reviewers’ acknowledgement of our rigorous evaluation design as well as the novelty of our SayNext-Chat framework. Your positive comments on our contributions are highly motivating for us.

We kindly encourage the reviewers to revisit the motivation of our work (lines 52–98), which already clarifies the conceptual significance of our SayNext. Below, we will further demonstrate the significance and the position of our work, which we have added to the revised manuscript (see Sec. 5, Limitations & Future Works).

**Significance**:

A central contribution of SayNext is that it aims to address a fundamental limitation of current LLMs: their inherently passive next-token prediction mechanism. Our preliminary experiments show that the classic LLMs generate responses by extrapolating statistical textual patterns, without modeling how humans actively anticipate others’ intentions during real interactions. Our framework takes a step toward a more cognitively grounded model of dialogue: integrating visual non-verbal cues, affective cues, and social context into a predictive-coding–inspired mechanism.

Although next-utterance prediction and traditional dialogue tasks share a similar surface form, their underlying cognitive objectives are fundamentally different. Dialogue modeling is a reactive (open-ended) task: the model simply produces a linguistically coherent and high-likelihood reply to another speaker, without needing to access or model that person’s internal state (seen in existing LLMs). Next-utterance prediction is anticipatory, which is a much more precise and advanced task: the model must predict what the same speaker will actually say next (linguistically coherent and high-likelihood reply is NOT enough), requiring precise inference of the speaker’s real intention, cognition, and emotional state from subtle multimodal cues. This shifts the task from language continuation to a more cognitively grounded reasoning process, making it conceptually distinct from traditional dialogue generation.

**Position of our paper**:

At a high level, our task is not a conventional dialogue-prediction problem (i.e., next-token prediction). Instead, our goal is to investigate a fundamental limitation in current LLMs and MLLMs: although these models excel at sustaining dialogue in traditional NLP settings, *they consistently fail to predict what a specific human will say next*. This gap is nontrivial and reflects a deeper issue: LLMs operate as passive next-token generators rather than systems capable of engaging in human-like anticipatory cognition. SayNext (next-utterance prediction) is therefore designed as a **proxy task** to model aspects of human predictive processing. Performing well on this task **implicitly** includes intention inference, affective anticipation, and multimodal social reasoning—capabilities that are broadly lacking in current LLMs. Consequently, we remind readers that the SayNext task does not fit neatly into existing categories such as emotion recognition, intention classification, or dialogue continuation. Instead, it opens a distinct and complementary research direction that we hope will motivate future work in cognitively grounded predictive modeling and inspire next-generation LLMs with more human-like anticipatory abilities.

As this is a pioneering work, we sincerely wish that reviewers can understand the difficulty and value of this SayNext benchmark, which brings new thoughts to the community, and we **hope the reviewers may reconsider the assessment in light of these clarifications.**

---

### Meta-Review · Area_Chair_CaUe · 2026-01-05

**Summary:**

The paper addresses a limitation of large language models (LLMs), namely, that they struggle to accurately predict next conversational utterances in human conversations.. The authors introduce a benchmark (SayNextBench) together with a dataset of real-world dialogues like post-match interviews (SayNext-PC). Moreover, they propose a cognitively inspired model (SayNext-Chat) that integrates verbal and non-verbal signals and uses “priming factors” to capture intent. In the experiments the  model outperforms state-of-the-art MLLMs in terms of lexical overlap, semantic similarity, and emotion consistency.

**Reviewer Concerns:**

A lack of novelty. The rebuttal argued that a large-scale dataset for  multimodal next-utterance prediction with continuous speaker focus does not exist so far.
The usage of GPT-4-1 for computing the priming factors. Authors added experiments with Llama and Gemini leading to comparable results.
All models do show low lexical overlap. The rebuttal justified BLEU/ROUGE as defensive baselines and emphasized that the proposed semantic, emotional, and human-preference metrics as more appropriate for this one-to-many prediction task.
Limited domain of the dataset (post-match tennis interviews and single turn dialogues). Authors argue that their cross-scenario experiments show good results on IEMOCAP as well and multi-turn dialogues are left for future work.
Unfair comparison between fine-tuned models and general LMMs. Authors pointed out that they already present one comparison to a fine-tuned LMM which still performs little worse than the proposed model. They also added an evaluation of the proposed priming approach in connection with another MLMM.
Missing justification of number of priming factores. The authors argue that this choice was data driven.
Limited analysis of failure cases and downstream impact. Authors added a failure case analysis, clarified downstream relevance, expanded ablations, and strengthened interpretability discussion, while acknowledging deeper cognitive validation as future work.

The rebuttal clarified most questions and presented several additional supportive results. Concens like the limited domain of the dataset and the keep existing. Those remaining concerns are primarily conceptual and about scope, rather than correctness. Given the borderline nature of the reviews and the lack of strong consensus that the contribution is sufficiently compelling at this stage, we regret that we are unable to recommend acceptance. We nevertheless believe this work is promising and encourage the authors to continue refining the task positioning and broadening its applicability.

**Reviewer Scores:**

Two reviewers voted for weak acceptance (6). I assume that these reviewers would have kept their scores after rebuttal. The other two voted for weak rejection (4). Their main concerns were primarily conceptual regarding the scope of the task and the limited geneality of the data set. Therefore, I guess the reviewers would not necessarily change their score after the rebuttal.

---

### Decision · Program_Chairs · 2026-01-26

Reject